# Learning sparse features can lead to overfitting in neural networks

**Leonardo Petrini** *
Institute of Physics
École Polytechnique Fédérale de Lausanne
leonardo.petrini@epfl.ch

**Francesco Cagnetta** *
Institute of Physics
École Polytechnique Fédérale de Lausanne
francesco.cagnetta@epfl.ch

**Eric Vanden-Eijnden**
Courant Institute of Mathematical Sciences
New York University
eve2@cims.nyu.edu

**Matthieu Wyart**
Institute of Physics
École Polytechnique Fédérale de Lausanne
matthieu.wyart@epfl.ch

## Abstract

It is widely believed that the success of deep networks lies in their ability to learn a meaningful representation of the features of the data. Yet, understanding when and how this feature learning improves performance remains a challenge: for example, it is beneficial for modern architectures trained to classify images, whereas it is detrimental for fully-connected networks trained on the same data. Here we propose an explanation for this puzzle, by showing that feature learning can perform worse than lazy training (via random feature kernel or the NTK) as the former can lead to a sparser neural representation. Although sparsity is known to be essential for learning anisotropic data, it is detrimental when the target function is constant or smooth along certain directions of input space. We illustrate this phenomenon in two settings: *(i)* regression of Gaussian random functions on the $d$-dimensional unit sphere and *(ii)* classification of benchmark datasets of images. For *(i)*, we compute the scaling of the generalization error with the number of training points and show that methods that do not learn features generalize better, even when the dimension of the input space is large. For *(ii)*, we show empirically that learning features can indeed lead to sparse and thereby less smooth representations of the image predictors. This fact is plausibly responsible for deteriorating the performance, which is known to be correlated with smoothness along diffeomorphisms.

## 1   Introduction

Neural networks are responsible for a technological revolution in a variety of machine learning tasks. Many such tasks require learning functions of high-dimensional inputs from a finite set of examples, thus should be generically hard due to the *curse of dimensionality* [1, 2]: the exponent that controls the scaling of the generalization error with the number of training examples is inversely proportional to the input dimension $d$. For instance, for standard image classification tasks with $d$ ranging in $10^3 \div 10^5$, such exponent should be practically vanishing, contrary to what is observed in practice [3]. In this respect, understanding the success of neural networks is still an open question. A popular explanation is that, during training, neurons adapt to features in the data that are relevant for the task [4], effectively reducing the input dimension and making the problem tractable [5, 6, 7]. However, understanding quantitatively if this intuition is true and how it depends on the structure of the task remains a challenge.

---

*Equal contribution (a coin was flipped).

36th Conference on Neural Information Processing Systems (NeurIPS 2022).

Figure 1: **Feature vs. Lazy in image classification.** Generalization error as a function of the training-set size $n$ for infinite-width fully-connected networks (FCNs) trained in the feature (blue) and lazy regime (orange). In the latter case the limit is taken exactly by training an SVC algorithm with the analytical NTK [23]. In the former case, the infinite-width limit can be accurately approximated for these datasets by considering very wide nets ($H = 10^3$), and performing ensemble averaging on different initial conditions of the parameters as shown in [24, 25]. Panels correspond to different benchmark image datasets [26, 27, 28]. Results are averaged over 10 different initializations of the networks and datasets.

Recently much progress was made in characterizing the conditions which lead to features learning, in the overparameterized setting where networks generally perform best. When the initialization scale of the network parameters is large [8] one encounters the *lazy training regime*, where neural networks behave as kernel methods [9, 10] (coined Neural Tangent Kernel or NTK) and features are not learned. By contrast, when the initialization scale is small, a *feature learning regime* is found [11, 12, 13] where the network parameters evolve significantly during training. This limit is much less understood apart from very simple architectures, where it can be shown to lead to sparse representations where a limited number of neurons are active after training [14]. Such sparse representations can also be obtained by regularizing the weights during training [2, 15].

In terms of performance, most theoretical works have focused on fully-connected networks. For these architectures, feature learning was shown to significantly outperform lazy training [16, 17, 18, 19, 11] for certain tasks, including approximating a function which depends only on a subset or a linear combination of the input variables. However, when such primitive networks are trained on image datasets, learning features is detrimental [20, 21], as illustrated in Fig. 1 (see [19, Fig. 3] for the analogous plot in the case of a target function depending on just one of the input variables, where learning features is beneficial). A similar result was observed in simple models of data [22]. These facts are unexplained, yet central to understanding the implicit bias of the feature learning regime.

## 1.1 Our contribution

Our main contribution is to provide an account of the drawbacks of learning sparse representations based on the following set of ideas. Consider, for concreteness, an image classification problem: *(i)* images class varies little along smooth deformations of the image; *(ii)* due to that, tasks like image classification require a continuous distribution of neurons to be represented; *(iii)* thus, requiring sparsity can be detrimental for performance. We build our argument as follows.

- In order to find a quantitative description of the phenomenon, we start from the problem of regression of a random target function of controlled smoothness on the $d$-dimensional unit sphere, and study the property of the minimizers of the empirical loss with $n$ observations, both in the lazy and the feature learning regimes. More specifically, we consider two extreme limits—the NTK limit and mean-field limit—as representatives of lazy and feature regimes, respectively (section 2). Both these limits admit a simple formulation that allows us to predict generalization performances. In particular, our results on feature learning rely on solutions having an atomic support. This property can be justified for one-hidden-layer neural networks with ReLU activations and weight decay. Yet, we also find such a sparsity empirically using gradient descent in the absence of regularization, if weights are initialized to be small enough.

- We find that lazy training leads to smoother predictors than feature learning. As a result, lazy training outperforms feature learning when the target function is also sufficiently smooth. Otherwise, the performances of the two methods are comparable, in the sense that they display the same asymptotic decay of generalization error with the number of training

examples. Our predictions are obtained from asymptotic arguments that we systematically back up with numerical studies.

- For image datasets, it is believed that diffeomorphisms of images are key transformations along which the predictor function should only mildly vary to obtain good performance [29]. From the results above, a natural explanation as to why lazy beats feature for fully connected networks is that it leads to predictors with smaller variations along diffeomorphisms. We confirm that this is indeed the case empirically on benchmark datasets.

Numerical experiments are performed in PyTorch [30], and the code for reproducing experiments is available online at github.com/pcsl-epfl/regressionsphere.

## 1.2  Related Work

The property that training ReLU networks in the feature regime leads to a sparse representation was observed empirically [31]. This property can be justified for one-hidden-layer networks by casting training as a L1 minimization problem [32, 2], then using a representer theorem [33, 15, 34]. This is analogous to what is commonly done in predictive sparse coding [35, 36, 37, 38].

Many works have investigated the benefits of learning sparse representations in neural networks. [2, 16, 17, 18, 19, 39, 40] study cases in which the true function only depends on a linear subspace of input space, and show that feature learning profitably capture such property. Even for more general problems, sparse representations of the data might emerge naturally during deep network training—a phenomenon coined *neural collapse* [41]. Similar sparsification phenomena, for instance, have been found to allow for learning convolutional layers from scratch [42, 43]. Our work builds on this body of literature by pointing out that learning sparse features can be detrimental, if the task does not allow for it.

There is currently no general framework to predict rigorously the learning curve exponent $\beta$ defined as $\epsilon(n) = \mathcal{O}(n^{-\beta})$ for kernels. Some of our asymptotic arguments can be obtained by other approximations, such as assuming that data points lie on a lattice in $\mathbb{R}^d$ [44], or by using the non-rigorous replica method of statistical physics [45, 46, 47]. In the case $d = 2$, we provide a more explicit mathematical formulation of our results, which leads to analytical results for certain kernels. We systematically back up our predictions with numerical tests as $d$ varies.

Finally, in the context of image classification, the connection between performance and 'stability' or smoothness toward small diffeomorphisms of the inputs has been conjectured by [29, 48]. Empirically, a strong correlation between these two quantities was shown to hold across various architectures for real datasets [49]. In that reference, it was found that fully connected networks lose their stability over training: here we show that this effect is much less pronounced in the lazy regime.

## 2  Problem and notation

**Task**  We consider a supervised learning scenario with $n$ training points $\{\boldsymbol{x}_i\}_{i=1}^n$ uniformly drawn on the $d$-dimensional unit sphere $\mathbb{S}^{d-1}$. We assume that the target function $f^*$ is an isotropic Gaussian random process on $\mathbb{S}^{d-1}$ and control its statistics via the spectrum: by introducing the decomposition of $f^*$ into spherical harmonics (see App. A for definitions),

$$f^*(\boldsymbol{x}) = \sum_{k \geq 0} \sum_{\ell=1}^{\mathcal{N}_{k,d}} f^*_{k,\ell} Y_{k,\ell}(\boldsymbol{x}) \quad \text{with} \quad \mathbb{E}\left[f^*_{k,\ell}\right] = 0, \quad \mathbb{E}\left[f^*_{k,\ell} f^*_{k',\ell'}\right] = c_k \delta_{k,k'} \delta_{\ell,\ell'}. \tag{2.1}$$

We assume that all the $c_k$ with $k$ odd vanish apart from $c_1$: this is required to guarantee that $f^*$ can be approximated as well as desired with a one-hidden-layer ReLU network with no biases, as discussed in App. A. We also assume that the non-zero $c_k$ decay as a power of $k$ for $k \gg 1$, $c_k \sim k^{-2\nu_t - (d-1)}$. The exponent $\nu_t > 0$ controls the (weak) differentiability of $f^*$ on the sphere (see App. A) and also the statistics of $f^*$ in real space:

$$\mathbb{E}\left[|f^*(\boldsymbol{x}) - f^*(\boldsymbol{y})|^2\right] = O\left(|\boldsymbol{x} - \boldsymbol{y}|^{2\nu_t}\right) = O\left((1 - \boldsymbol{x} \cdot \boldsymbol{y})^{\nu_t}\right) \quad \text{as} \quad \boldsymbol{x} \to \boldsymbol{y}. \tag{2.2}$$

Examples of such a target function for $d = 3$ and different values of $\nu_t$ are reported in Fig. 2.

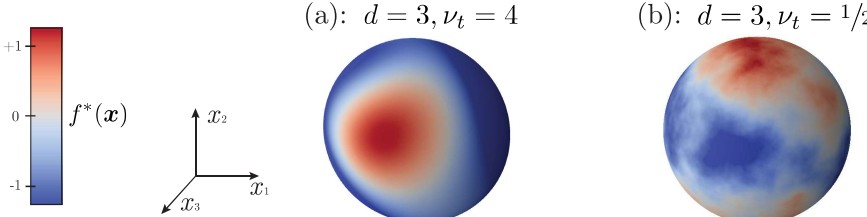

Figure 2: **Gaussian random process on the sphere.** We show here two samples of the task introduced in section 2 when the target function $f^*(\boldsymbol{x})$ is defined on the $3-$dimensional unit sphere. (a) and (b) show samples of large and small smoothness coefficient $\nu_t$, respectively.

**Neural network representation in the feature regime** In this regime we aim to approximate the target function $f^*(x)$ via a *one-hidden-layer neural network* of width $H$,

$$f_H(\boldsymbol{x}) = \frac{1}{H} \sum_{h=1}^{H} w_h \sigma(\boldsymbol{\theta}_h \cdot \boldsymbol{x}), \tag{2.3}$$

where $\{\boldsymbol{\theta}_h\}_{h=1}^{H}$ (the features) and $\{w_h\}_{h=1}^{H}$ (the weights) are the network parameters to be optimized, and $\sigma(x)$ denotes the ReLU function, $\sigma(x) = \max\{0, x\}$. If we assume that $\{\boldsymbol{\theta}_h, w_h\}_{h=1}^{H}$ are independently drawn from a probability measure $\mu$ on $\mathbb{S}^{d-1} \times \mathbb{R}$ such that the Radon measure $\gamma = \int_{\mathbb{R}} w\mu(\cdot, dw)$ exists, then as $H \to \infty$,

$$\lim_{H\to\infty} f_H(\boldsymbol{x}) = \int_{\mathbb{S}^{d-1}} \sigma(\boldsymbol{\theta} \cdot \boldsymbol{x}) d\gamma(\boldsymbol{\theta}) \qquad \text{a.e. on } \mathbb{S}^{d-1}. \tag{2.4}$$

This is the so-called mean-field limit [11, 12], and it is then natural to determine the optimal $\gamma$ via

$$\gamma^* = \arg\min_{\gamma} \int_{\mathbb{S}^{d-1}} |d\gamma(\boldsymbol{\theta})| \quad \text{subject to:} \quad \int_{\mathbb{S}^{d-1}} \sigma(\boldsymbol{\theta} \cdot \boldsymbol{x}_i) d\gamma(\boldsymbol{\theta}) = f^*(\boldsymbol{x}_i) \quad \forall i = 1, \ldots, n. \tag{2.5}$$

In practice, we can approximate this minimization problem by using a network with large but finite width, constraining the feature to be on the sphere $|\boldsymbol{\theta}_h| = 1$, and minimizing the following empirical loss with L1 regularization on the weights,

$$\min_{\substack{\{w_h, \boldsymbol{\theta}_h\}_{h=1}^{H} \\ |\boldsymbol{\theta}_h|=1}} \frac{1}{2n} \sum_{i=1}^{n} \left( f^*(\boldsymbol{x}_i) - \frac{1}{H} \sum_{h=1}^{H} w_h \sigma(\boldsymbol{\theta}_h \cdot \boldsymbol{x}_i) \right)^2 + \frac{\lambda}{H} \sum_{h=1}^{H} |w_h|. \tag{2.6}$$

This minimization problem leads to (2.5) when $H \to \infty$ and $\lambda \to 0$. Note that, by homogeneity of ReLU, (2.6) can be shown to be equivalent to imposing a regularization on the L2 norm of all parameters [32, Thm. 10], i.e. the usual weight decay.

To proceed we will make the following assumption about the minimizer $\gamma^*$:

**Assumption 1.** *The minimizer $\gamma^*$ of (2.5) is unique and atomic, with $n_A \leq n$ atoms, i.e. there exists $\{w_i^*, \boldsymbol{\theta}_i^*\}_{i=1}^{n_A}$ such that*

$$\gamma^* = \sum_{i=1}^{n_A} w_i^* \delta_{\boldsymbol{\theta}_i^*}. \tag{2.7}$$

The main component of the assumption is the uniqueness of $\gamma^*$; if it holds the sparsity of $\gamma^*$ follows from the representer theorem, see e.g. [33]. Both the uniqueness and sparsity of the minimizer can be justified as holding generically using asymptotic arguments involving recasting the $L1$ minimization problem 2.5 as a linear programming one: these arguments are standard (see e.g. [50]) and are presented in App. B for the reader convenience. In our arguments below to deduce the scaling of the generalization error we will mainly use that $n_A = O(n)$—we shall confirm this fact numerically even in the absence of regularization, if the weights are initialized to be small enough. Notice that from Assumption 1 it follows that the predictor in the feature regime corresponding to the minimizer $\gamma^*$ takes the following form

$$f^{\text{FEATURE}}(\boldsymbol{x}) = \sum_{i=1}^{n_A} w_i^* \sigma(\boldsymbol{\theta}_i^* \cdot \boldsymbol{x}). \tag{2.8}$$

**Neural network representation in the lazy regime.** In this regime we approximate the target function $f^*(x)$ via

$$f^{\text{NTK}}(\boldsymbol{x}) = \sum_{i=1}^{n} g_i K^{\text{NTK}}(\boldsymbol{x}_i \cdot \boldsymbol{x}), \tag{2.9}$$

where the weights $\{g_i\}_{i=1}^n$ solve

$$f^*(\boldsymbol{x}_j) = \sum_{i=1}^{n} g_i K^{\text{NTK}}(\boldsymbol{x}_i \cdot \boldsymbol{x}_j), \qquad j = 1, \ldots, n. \tag{2.10}$$

and $K^{\text{NTK}}(\boldsymbol{x} \cdot \boldsymbol{y})$ is the *Neural Tangent Kernel* (NTK) [9]

$$K^{\text{NTK}}(\boldsymbol{x} \cdot \boldsymbol{y}) = \int_{\mathbb{S}^{d-1} \times \mathbb{R}} \left( \sigma(\boldsymbol{\theta} \cdot \boldsymbol{x})\sigma(\boldsymbol{\theta} \cdot \boldsymbol{y}) + w^2 \, \boldsymbol{x} \cdot \boldsymbol{y} \, \sigma'(\boldsymbol{\theta} \cdot \boldsymbol{x})\sigma'(\boldsymbol{\theta} \cdot \boldsymbol{y}) \right) d\mu_0(\boldsymbol{\theta}, w). \tag{2.11}$$

Here $\mu_0$ is a fixed probability distribution which, in the NTK training regime [9], is the distribution of the features and weights at initialization. It is well-known [51] that the solution to kernel ridge regression problem can also be expressed via the kernel trick as

$$f^{\text{NTK}}(\boldsymbol{x}) = \int_{\mathbb{S}^{d-1} \times \mathbb{R}} \left( g_w(\boldsymbol{\theta}, w)\sigma(\boldsymbol{\theta} \cdot \boldsymbol{x}) + w\boldsymbol{x} \cdot \boldsymbol{g}_\theta(\boldsymbol{\theta}, w)\sigma'(\boldsymbol{\theta} \cdot \boldsymbol{x}) \right) d\mu_0(\boldsymbol{\theta}, w) \tag{2.12}$$

where $\boldsymbol{g}_\theta$ and $g_w$ are the solutions of

$$\min_{g_w, \boldsymbol{g}_\theta} \int_{\mathbb{S}^{d-1} \times \mathbb{R}} \left( g_w^2(w, \boldsymbol{\theta}) + |\boldsymbol{g}_\theta(w, \boldsymbol{\theta})|^2 \right) d\mu_0(\boldsymbol{\theta}, w)$$

$$\text{subject to:} \int_{\mathbb{S}^{d-1} \times \mathbb{R}} \left( g_w(w, \boldsymbol{\theta})\sigma(\boldsymbol{\theta} \cdot \boldsymbol{x}_i) + w\boldsymbol{x}_i \cdot \boldsymbol{g}_\theta(w, \boldsymbol{\theta})\sigma'(\boldsymbol{\theta} \cdot \boldsymbol{x}_i) \right) d\mu_0(\boldsymbol{\theta}, w) = f^*(\boldsymbol{x}_i) \tag{2.13}$$

$$\forall i = 1, \ldots, n.$$

Another lazy limit can be obtained equivalently by training only the weights while keeping the features to their initialization value. This is equivalent to forcing $\boldsymbol{g}_\theta(\boldsymbol{\theta}, w)$ to vanish in Eq. 2.13, resulting again in a kernel method. The kernel, in this case, is called *Random Feature Kernel* ($K^{\text{RFK}}$), and can be obtained from Eq. 2.11 by setting $d\mu_0(\boldsymbol{\theta}, w) = \delta_{w=0}d\tilde{\mu}_0(\boldsymbol{\theta})$. The minimizer can then be written as in Eq. 2.9 with $K^{\text{NTK}}$ replaced by $K^{\text{RFK}}$.

## 3 Asymptotic analysis of generalization

In this section, we characterize the asymptotic decay of the generalization error $\bar{\epsilon}(n)$ averaged over several realizations of the target function $f^*$. Denoting with $d\tau^{d-1}(\boldsymbol{x})$ the uniform measure on $\mathbb{S}^{d-1}$,

$$\bar{\epsilon}(n) = \mathbb{E}_{f^*} \left[ \int d\tau^{d-1}(\boldsymbol{x}) \, (f^n(\boldsymbol{x}) - f^*(\boldsymbol{x}))^2 \right] = \mathcal{A}_d n^{-\beta} + o(n^{-\beta}), \tag{3.1}$$

for some constant $\mathcal{A}_d$ which might depend on $d$ but not on $n$. Both for the lazy (see Eq. 2.9) and feature regimes (see Eq. 2.8) the predictor can be written as a sum of $\mathcal{O}(n)$ terms:

$$f^n(\boldsymbol{x}) = \sum_{j=1}^{\mathcal{O}(n)} g_j \varphi(\boldsymbol{x} \cdot \boldsymbol{y}_j) := \int_{\mathbb{S}^{d-1}} g^n(\boldsymbol{y})\varphi(\boldsymbol{x} \cdot \boldsymbol{y})d\tau(\boldsymbol{y}). \tag{3.2}$$

In the feature regime, the $g_j$'s ($\boldsymbol{y}_j$) coincide with the optimal weights $w_j^*$ (features $\boldsymbol{\theta}_j^*$), $\varphi$ with the activation function $\sigma$. In the lazy regime, the $\boldsymbol{y}_j$ are the training points $\boldsymbol{x}_j$, $\varphi$ is the neural tangent or random feature kernel the $g_j$'s are the weights solving Eq. 2. We have defined the density $g^n(\boldsymbol{x}) = \sum_j |\mathbb{S}^{d-1}|g_j\delta(\boldsymbol{x} - \boldsymbol{y}_j)$ so as to cast the predictor as a convolution on the sphere. Therefore, the projections of $f^n$ onto spherical harmonics $Y_{k,\ell}$ read $f_{k,\ell}^n = g_{k,\ell}^n \varphi_k$, where $g_{k,\ell}^n$ is the projection of $g^n(\boldsymbol{x})$ and $\varphi_k$ that of $\varphi(\boldsymbol{x} \cdot \boldsymbol{y})$. For ReLU neurons one has (as shown in App. A)

$$\varphi_k^{\text{LAZY}} \sim k^{-(d-1)-2\nu} \quad \text{with } \nu = 1/2 \text{ (NTK)}, 3/2 \text{ (RFK)}, \quad \varphi_k^{\text{FEATURE}} \sim k^{-\frac{d-1}{2}-3/2}. \tag{3.3}$$

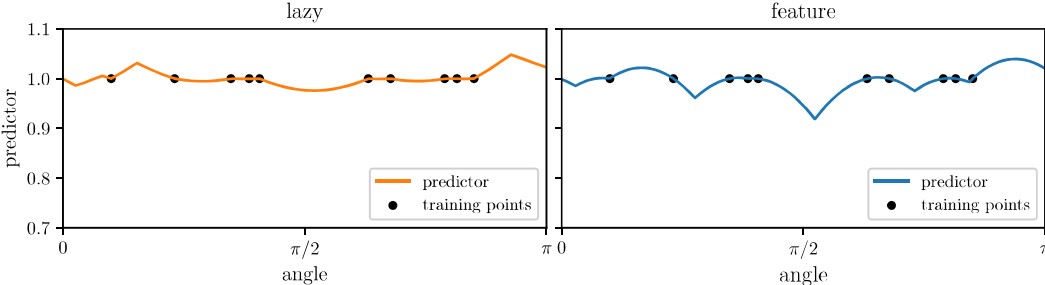

Figure 3: **Feature vs. Lazy Predictor.** Predictor of the lazy (left) and feature (right) regime when learning the constant function on the ring with 8 uniformly-sampled training points.

**Main Result** Consider a target function $f^*$ with smoothness exponent $\nu_t$ as defined above, with data lying on $\mathbb{S}^{d-1}$. If $f^*$ is learnt with a one-hidden-layer network with ReLU neurons in the regimes specified above, then the generalization error follows $\bar{\epsilon}(n) \sim n^{-\beta}$ with:

$$\beta^{\text{LAZY}} = \frac{\min\left\{2(d-1) + 4\nu, 2\nu_t\right\}}{d-1} \text{ with } \nu = \begin{cases} 1/2 \text{ for NTK,} \\ 3/2 \text{ for RFK,} \end{cases}, \tag{3.4a}$$

$$\beta^{\text{FEATURE}} = \frac{\min\left\{(d-1) + 3, 2\nu_t\right\}}{d-1}. \tag{3.4b}$$

This is our central result. It implies that if the target function is a smooth isotropic Gaussian field (realized for large $\nu_t$), then lazy beats feature, in the sense that training the network in the lazy regime leads to a better scaling of the generalization performance with the number of training points.

**Strategy** There is no general framework for a rigorous derivation of the generalization error in the ridgeless limit $\lambda \to 0$: predictions such as that of Eq. 3.4 can be obtained by either assuming that training points (for Eq. 3.4a) and neurons (for Eq. 3.4b) lie on a periodic lattice [44], or (for Eq. 3.4a) using the replica method from physics [45] as shown in App. F. Here we follow a different route, by first characterizing the form of the predictor for $d = 2$ (proof in App. C). This property alone allows us to determine the asymptotic scaling of the generalization error. We use it to analytically obtain the generalization error in the NTK case with a slightly simplified function $\varphi$ (details in App. D). This calculation motivates a simple ansatz for the form of $g^n(\boldsymbol{x})$ entering Eq. 3.2 and its projections onto spherical harmonics, which extends naturally to arbitrary dimension. We confirm the predictions resulting from this ansatz systematically in numerical experiments.

**Properties of the predictor in $d = 2$** On the unit circle $\mathbb{S}^1$ all points are identified by a polar angle $x \in [0, 2\pi)$. Hence both target function and estimated predictor are functions of the angle, and all functions of a scalar product are in fact functions of the difference in angle. In particular, introducing $\tilde{\varphi}(x) = \varphi(\cos(x))$,

$$f^n(x) = \sum_j g_j \tilde{\varphi}(x - x_j) \equiv \int_0^{2\pi} \frac{dy}{2\pi} g^n(y) \tilde{\varphi}(x - y), \tag{3.5}$$

where we defined

$$g^n(x) = \sum_{j=1}^n (2\pi g_j) \delta(y - x_j). \tag{3.6}$$

Both for feature regime and NTK limit, the first derivative of $\tilde{\varphi}(x)$ is continuous except for two values of $x$ (0 and $\pi$ for lazy, $-\pi/2$ and $\pi/2$ for feature), so that $\tilde{\varphi}(x)''$ has a singular part consisting of two Dirac delta functions.

As a result, the second derivative of the predictor $(f^n)''$ has a singular part consisting of many Dirac deltas. If we denote with $(f^n)''_r$ the regular part, obtained by subtracting all the delta functions, we can show that (see App. C):

**Proposition 1.** *(informal) As $n \to \infty$, $(f^n)''_r$ converges to a function having finite second moment, i.e.*

$$\lim_{n \to \infty} \mathbb{E}_{f^*}[(f^n)''_r(x)]^2 = const. < \infty. \tag{3.7}$$

In the large $n$ limit, the predictor displays a singular second derivative at $O(n)$ points. Proposition 1 implies that outside of these singular points the second derivative is well defined. Thus, as $n$ gets large and the singular points approach each other, the predictor can be approximated by a chain of parabolas, as highlighted in Fig. 3 and noticed in [47] for a Laplace kernel. This property alone allows to determine the asymptotic scaling of the error in $d = 2$. In simple terms, Prop. 1 follows from the convergence of $g^n$ to the function satisfying $f^*(x) = \int \frac{dy}{2\pi} g(y) \tilde{\varphi}_r(x - y)$, which is guaranteed under our assumptions on the target function—a detailed proof is given in App. C.

**Decay of the error in** $d = 2$ **(sketch)** The full calculation is in App. D. Consider a slightly simplified problem where $\tilde{\varphi}$ has a single discontinuity in its derivative, located at $x = 0$. In this case, $f^n(x)$ is singular if and only if $x$ is a data point. Consider then the interval $x \in [x_i, x_{i+1}]$ and set $\delta_i = x_{i+1} - x_i$, $x_{i+1/2} = (x_{i+1} + x_i)/2$. If the target function is smooth enough ($\nu_t > 2$), then a Taylor expansion implies $|f^*(x_{i+1/2}) - f^n(x_{i+1/2})| \sim \delta_i^2$. Since the distances $\delta_i$ between adjacent singular points are random variables with mean of order $1/n$ and finite moments, it is straightforward to obtain that $\bar{\epsilon}(n) \sim \sum_i (f^*(x_{i+1/2}) - f^n(x_{i+1/2}))^2 \sim \sum_i \delta_i^4 \sim n^{-4}$. By contrast if $f^*$ is not sufficiently smooth ($\nu_t \leq 2$), then $|f^*(x_{i+1/2}) - f^n(x_{i+1/2})| \sim \delta_i^{2\nu_t}$, leading to $\bar{\epsilon}(n) \sim n^{-2\nu_t}$. Note that for this asymptotic argument to apply to the feature learning regime, one must ensure that the distribution of the rescaled distance between adjacent singularities $n\delta_i$ has a finite fourth moment. This is obvious in the lazy regime, where the $\delta_i$'s are controlled by the position of the training points, but not in the feature regime, where the distribution of singular points is determined by that of the neuron's features. Nevertheless, we show that it must be the case in our setup in App. D.

**Interpretation in terms of spectral bias** From the discussion above it is evident that there is a length scale $\delta$ of order $1/n$ such that $f^n(x)$ is a good approximation of $f^*(x)$ over scales larger than $\delta$. In terms of Fourier modes[2], one has: *i)* $\widehat{f^n}(k)$ matches $\widehat{f^n}(k)$ at long wavelengths, i.e. for $k \ll k_c \sim 1/n$. *ii)* In addition, since the phases $\exp(ikx_j)$ become effectively random phases for $k \gg k_c$, $\widehat{g^n}(k) = \sum_j g_j \exp(ikx_j)$ becomes a Gaussian random variable with zero mean and fixed variance and thus *iii)* $\widehat{f^n}(k) = \widehat{g^n}(k)\widehat{\varphi}(k)$ decorrelates from $f^*$ for $k \gg k_c$. Therefore

$$\bar{\epsilon}(n) \sim \sum_{|k| > k_c} \mathbb{E}_{f^*}\left[\left(\widehat{g^n}(k)\widehat{\varphi}(k) - \widehat{f^n}(k)\right)^2\right] \sim \sum_{|k| \geq k_c} \mathbb{E}_{f^*}\left[(\widehat{g^n}(k))^2\right]\widehat{\varphi}(k)^2 + \mathbb{E}_{f^*}\left[(\widehat{f^n}(k))^2\right].$$
(3.8)

For $\nu_t > 2$, one has $\sum_j g_j^2 \sim n^{-1}\lim_{n\to\infty}\int g^n(x)^2 dx \sim n^{-1}$. It follows (see App. E for details) that the sum is dominated by the first term, hence entirely controlled by the Fourier coefficients of $\widehat{f^n}(k)$ at large $k$. A smoother predictor corresponds to a faster decay of $\widehat{f^n}(k)$ with $k$, thus a faster decay of the error with $n$. Plugging the relevant decays yields $\bar{\epsilon} \sim n^{-4}$ for feature regime and lazy regime with the NTK, and $n^{-6}$ for lazy regime with the RFK (which is smoother than the NTK). For $\nu_t \leq 2$, the two terms have comparable magnitude (see App. E), thus $\bar{\epsilon} \sim n^{-2\nu_t}$.

**Generalization to higher dimensions** The argument above can be generalized for any $d$ by replacing Fourier modes with projections onto spherical harmonics. The characteristic distance between training points scales as $n^{-1/(d-1)}$, thus $k_c \sim n^{-1/(d-1)}$. Our ansatz is that, as in $d = 2$: *i)* for $k \ll k_c$, the predictor modes coincide with those of the target function, $f^n_{k,l} \approx f^*_{k,l}$ (this corresponds to the spectral bias result of kernel methods, stating that the predictor reproduces the first $O(n)$ projections of the target in the kernel eigenbasis [45]); *ii)* For $k \gg k_c$, $g^n_{k,l}$ is a sum of uncorrelated terms, thus a Gaussian variable with zero mean and fixed variance; *iii)* $f^n_{k,\ell} = g^n_{k,\ell}\tilde{\varphi}_k$ decorrelates from $f^*_{k,\ell}$ for $k \gg k_c$. *i)*, *ii)* and *iii)* imply that:

$$\bar{\epsilon}(n) \sim \sum_{k \geq k_c}\sum_{l=1}^{\mathcal{N}_{k,d}} \mathbb{E}_{f^*}\left[\left(f^n_{k,l} - f^*_{k,l}\right)^2\right] \sim \sum_{k \geq k_c}\sum_{l=1}^{\mathcal{N}_{k,d}} \mathbb{E}_{f^*}\left[(g^n_{k,l})^2\right]\varphi_k^2 + k^{-2\nu_t-(d-1)}.$$
(3.9)

As shown in App. E, from this expression it is straightforward to obtain Eq. 3.4. Notice again that when the target is sufficiently smooth so that the predictor-dependent term dominates, the error is determined by the smoothness of the predictor. In particular, as $d > 2$, the predictor of feature learning is less smooth than both the NTK and RFK ones, due to the slower decay of the corresponding $\varphi_k$.

---

[2] The Fourier transform of a function $f(x)$ is indicated by the hat, $\widehat{f}(k)$.

## 4 Numerical tests of the theory

We test successfully our predictions by computing the learning curves of both lazy and feature regimes when *(i)* the target function is constant on the sphere for varying $d$, see Fig. 4, and *(ii)* the target is a Gaussian random field with varying smoothness $\nu_t$, as shown in Fig. G.1 of App. G. For the lazy regime, we perform kernel regression using the analytical expression of the NTK [52] (see also Eq. A.19). For the feature regime, we find that our predictions hold when having a small regularization, although it takes unreachable times for gradient descent to exactly recover the minimal-norm solution—a more in-depth discussion can be found in App. G. An example of the atomic distribution of neurons found after training, which contrasts with the initial distribution, is displayed in Fig. 5a, left panel.

Another way to obtain sparse features is to initialize the network with very small weights [14], as proposed in [8]. As in the presence of an infinitesimal weights decay, this scheme also leads to sparse solutions with $n_A = \mathcal{O}(n)$ – an asymptotic dependence confirmed in Fig. G.3 of App. G. This observation implies that our predictions must apply in that case too, as we confirm in Fig. G.3.

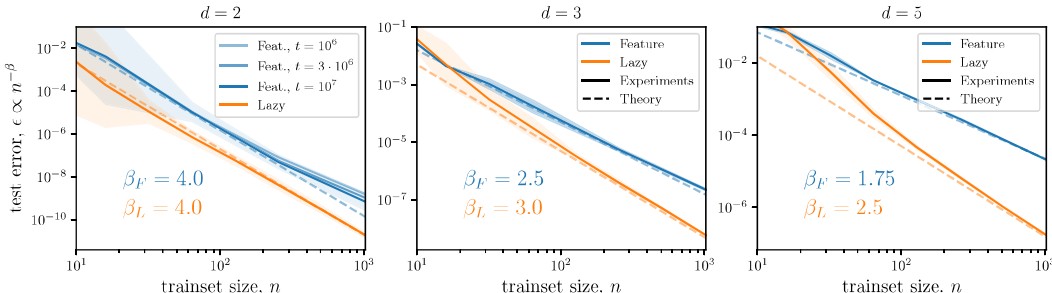

Figure 4: **Generalization error for a constant function** $f^*(\boldsymbol{x}) = 1$. Generalization error as a function of the training set size $n$ for a network trained in the feature regime with L1 regularization (blue) and kernel regression corresponding to the infinite-width lazy regime (orange). Numerical results (full lines) and the exponents predicted by the theory (dashed) are plotted. Panels correspond to different input-space dimensions ($d = 2, 3, 5$). Results are averaged over 10 different initializations of the networks and datasets. For $d = 2$ and large $n$, the gap between experiments and prediction for the feature regime is due to the finite training time $t$. Indeed our predictions become more accurate as $t$ increases, as illustrated in the left.

## 5 Evidence for overfitting along diffeomorphisms in image datasets

For fully-connected networks, the feature regime is well-adapted to learn anisotropic tasks [16]: if the target function does not depend on a certain linear subspace of input space, e.g. the pixels at the corner of an image, then neurons align perpendicularly to these directions [19]. By contrast, our results highlight a drawback of this regime when the target function is constant or smooth along directions in input space that require a continuous distribution of neurons to be represented. In such a case, the adaptation of the weights to the training points leads to a predictor with a sparse representation. Such a predictor would be less smooth than in the lazy regime and thus underperform.

Does this view hold for images, and explain why learning their features is detrimental for fully-connected networks? The first positive empirical evidence is that the neurons' distribution of networks trained on image data becomes indeed sparse in the feature regime, as illustrated in Fig. 5a, right, for CIFAR10 [28]. This observation raises the question of which are the directions in input space *i)* along which the target should vary smoothly, and *ii)* that are not easily represented by a discrete set of neurons. An example of such directions are global translations, which conserve the norm of the input and do not change the image class: the lazy regime predictor is indeed smoother than the feature one with respect to translations of the input (see App. H). Yet, these transformations live in a space of dimension 2, which is small in comparison with the full dimensionality $d$ of the data and thus may play a negligible role.

A much larger class of transformations believed to have little effect on the target are small diffeomorphisms [29]. A diffeomorphism $\tau$ acting on an image is illustrated in Fig. 5b, which highlights that

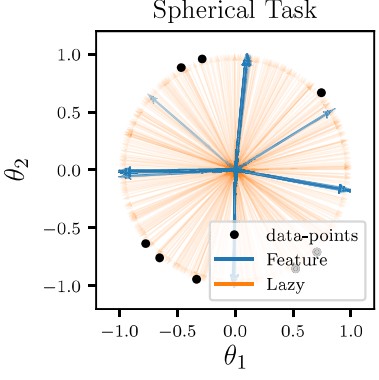 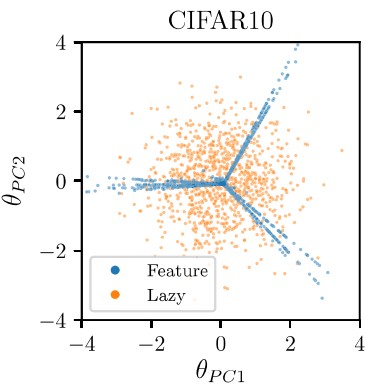 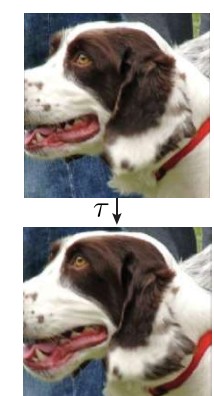

(a) **Features sparsification.** 1$^{st}$Panel: Distribution of neuron's feature for the task of learning a constant function on the sphere in 2D. Arrows represent a subset of the network features $\{\boldsymbol{\theta}_h\}_{h=1}^H$ after training in the lazy and feature regimes. Training is performed on $n = 8$ data-points (black dots). 2$^{nd}$Panel: FCN trained on CIFAR10. On the axes the first two principal components of the features $\{\boldsymbol{\theta}_h\}_{h=1}^H$ after training on $n = 32$ points in the feature (blue) and lazy (orange) regimes. Similarly to what is observed when learning a constant function, the $\boldsymbol{\theta}_h$ angular distribution becomes sparse with training in the feature regime.

(b) **Example of diffeomorphism.** Sample of a max-entropy deformation $\tau$ [49] when applied to a natural image, illustrating that it does not change the image class for the human brain.

Figure 5: **Features sparsification and example of a diffeomorphism.**

our brain still perceives the content of the transformed image as in the original one. Near-invariance of the task to these transformations is believed to play a key role in the success of deep learning, and in explaining how neural networks beat the curse of dimensionality [48]. Indeed, if modern architectures can become insensitive to these transformations, then the dimensionality of the problem is considerably reduced. In fact, it was found that the architectures displaying the best performance are precisely those which learn to vary smoothly along such transformations [49].

Small diffeomorphisms are likely the directions we are looking for. To test this hypothesis, following [49], we characterize the smoothness of a function along such diffeomorphisms, relative to that of random directions in input space. Specifically, we use the *relative sensitivity*:

$$R_f = \frac{\mathbb{E}_{x,\tau}\|f(\tau x) - f(x)\|^2}{\mathbb{E}_{x,\eta}\|f(x + \eta) - f(x)\|^2}. \tag{5.1}$$

In the numerator, the average is made over the test set and over an ensemble of diffeomorphisms, reviewed in App. I. The magnitude of the diffeomorphisms is chosen so that each pixel is shifted by one on average. In the denominator, the average runs over the test set and the vectors $\eta$ sampled uniformly on the sphere of radius $\|\eta\| = \mathbb{E}_{x,\tau}\|\tau x - x\|$, and this fixes the transformations magnitude.

We measure $R_f$ as a function of $n$ for three benchmark datasets of images, as shown in Fig. 6. We indeed find that $R_f$ is consistently smaller in the lazy training regime, where features are not learned. Overall, this observation supports the view that learning sparse features is detrimental when data present (near) invariance to transformations that cannot be represented sparsely by the architecture considered. Fig. 1 supports the idea that—for benchmark image datasets—this negative effect overcomes well-known positive effects of learning features, e.g. becoming insensitive to pixels on the edge of images (see App. H for evidence of this effect).

## 6   Conclusion

Our central result is that learning sparse features can be detrimental if the task presents invariance or smooth variations along transformations that are not adequately captured by the neural network architecture. For fully-connected networks, these transformations can be rotations of the input, but also continuous translations and diffeomorphisms.

Figure 6: **Sensitivity to diffeomorphisms vs number of training points.** Relative sensitivity of the predictor to small diffeomorphisms of the input images, in the two regimes, for varying number of training points $n$ and different image datasets. Smaller values correspond to a smoother predictor, on average. Results are computed using the same predictors as in Fig. 1.

Our analysis relies on the sparsity of the features learned by a shallow fully-connected architecture: even in the infinite width limit, when trained in the feature learning regime such networks behave as $\mathcal{O}(n)$ neurons. The asymptotic analysis we perform for random Gaussian fields on the sphere leads to predictions for the learning curve exponent $\beta$ in different training regimes, which we verify. Such kind of results is scarce in the literature.

Note that our analysis focuses on ReLU neurons because *(i)* these are very often used in practice and *(ii)* in that case, $\beta$ will depend on the training regime, allowing for stringent numerical tests. If smooth activations (e.g. softplus) are considered, we expect that learning features will still be detrimental for generalization. Yet, the difference will not appear in the exponent $\beta$, but in other aspects of the learning curves (including numerical coefficients and pre-asymptotic effects) that are harder to predict.

Most fundamentally, our results underline that the success of feature learning for modern architectures still lacks a sufficient explanation. Indeed, most of the theoretical studies that previously emphasized the benefits of learning features have been considering fully-connected networks, for which learning features can be in practice a drawback. It is tempting to argue that in modern architectures, learning features is not at a disadvantage because smoothness along diffeomorphisms can be enforced from the start—due to the locally connected, convolutional, and pooling layers [53, 29]. Yet the best architectures often do not perform pooling and are not stable toward diffeomorphisms at initialization. *During training*, learning features leads to more stable and smoother solutions along diffeomorphisms [54, 49]. Understanding why building sparse features enhances stability in these architectures may ultimately explain the magical feat of deep CNNs: learning tasks in high dimensions.

# Acknowledgements

We thank Lénaïc Chizat, Antonio Sclocchi, and Umberto M. Tomasini for helpful discussions. The work of MW is supported by a grant from the Simons Foundation (#454953) and from the NSF under Grant No. 200021-165509. The work of EVE is supported by the National Science Foundation under awards DMR-1420073, DMS-2012510, and DMS-2134216, by the Simons Collaboration on Wave Turbulence, Grant No. 617006, and by a Vannevar Bush Faculty Fellowship.

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
