# A    Quick recap of spherical harmonics

**Spherical harmonics**    This appendix collects some introductory background on spherical harmonics and dot-product kernels on the sphere [55]. See [56, 57] for an expanded treatment. Spherical harmonics are homogeneous polynomials on the sphere $\mathbb{S}^{d-1} = \{\boldsymbol{x} \in \mathbb{R}^d \mid \|\boldsymbol{x}\| = 1\}$, with $\|.\|$ denoting the L2 norm. Given the polynomial degree $k \in \mathbb{N}$, there are $\mathcal{N}_{k,s}$ linearly independent spherical harmonics of degree $k$ on $\mathbb{S}^{s-1}$, with

$$\mathcal{N}_{k,d} = \frac{2k+d-2}{k}\binom{d+k-3}{k-1}, \quad \begin{cases} \mathcal{N}_{0,d} = 1 \quad \forall d, \\ \mathcal{N}_{k,d} \asymp A_d k^{d-2} \quad \text{for } k \gg 1, \end{cases} \tag{A.1}$$

where $\asymp$ means logarithmic equivalence for $k \to \infty$ and $A_d = \sqrt{2/\pi}(d-2)^{\frac{3}{2}-d}e^{d-2}$. Thus, we can introduce a set of $\mathcal{N}_{k,d}$ spherical harmonics $Y_{k,\ell}$ for each $k$, with $\ell$ ranging in $1, \dots, \mathcal{N}_{k,d}$, which are orthonormal with respect to the uniform measure on the sphere $d\tau(\boldsymbol{x})$,

$$\{Y_{k,\ell}\}_{k \geq 0, \ell = 1, \dots, \mathcal{N}_{k,d}}, \quad \langle Y_{k,\ell}, Y_{k,\ell'} \rangle_{\mathbb{S}^{d-1}} := \int_{\mathbb{S}^{d-1}} Y_{k,\ell}(\boldsymbol{x}) Y_{k,\ell'}(\boldsymbol{x}) \, d\tau(\boldsymbol{x}) = \delta_{\ell,\ell'}. \tag{A.2}$$

Because of the orthogonality of homogeneous polynomials with different degree, the set is a complete orthonormal basis for the space of square-integrable functions on $\mathbb{S}^{d-1}$. For any function $f: \mathbb{S}^{d-1} \to \mathbb{R}$, then

$$f(\boldsymbol{x}) = \sum_{k \geq 0} \sum_{\ell=1}^{\mathcal{N}_{k,d}} f_{k,\ell} Y_{k,\ell}(\boldsymbol{x}), \quad f_{k,\ell} = \int_{\mathbb{S}^{d-1}} f(\boldsymbol{x}) Y_{k,\ell}(\boldsymbol{x}) d\tau(\boldsymbol{x}). \tag{A.3}$$

Furthermore, spherical harmonics are eigenfunctions of the Laplace-Beltrami operator $\Delta$, which is nothing but the restriction of the standard Laplace operator to $\mathbb{S}^{d-1}$,

$$\Delta Y_{k,\ell} = -k(k+d-2) Y_{k,\ell}. \tag{A.4}$$

**Legendre polynomials**    By fixing a direction $\boldsymbol{y}$ in $\mathbb{S}^{d-1}$ one can select, for each $k$, the only spherical harmonic of degree $k$ which is invariant for rotations that leave $\boldsymbol{y}$ unchanged. This particular spherical harmonic is, in fact, a function of $\boldsymbol{x} \cdot \boldsymbol{y}$ and is called the Legendre polynomial of degree $k$, $P_{k,d}(\boldsymbol{x} \cdot \boldsymbol{y})$ (also referred to as Gegenbauer polynomial). Legendre polynomials can be written as a combination of the orthonormal spherical harmonics $Y_{k,\ell}$ via the addition theorem [56, Thm. 2.9],

$$P_{k,d}(\boldsymbol{x} \cdot \boldsymbol{y}) = \frac{1}{\mathcal{N}_{k,d}} \sum_{\ell=1}^{\mathcal{N}_{k,d}} Y_{k,\ell}(\boldsymbol{x}) Y_{k,\ell}(\boldsymbol{y}). \tag{A.5}$$

Alternatively, $P_{k,d}$ is given explicitly as a function of $t = \boldsymbol{x} \cdot \boldsymbol{y} \in [-1, 1]$ via the Rodrigues' formula [56, Thm. 2.23],

$$P_{k,d}(t) = \left(-\frac{1}{2}\right)^k \frac{\Gamma\left(\frac{d-1}{2}\right)}{\Gamma\left(k+\frac{d-1}{2}\right)} \left(1-t^2\right)^{\frac{3-d}{2}} \frac{d^k}{dt^k} \left(1-t^2\right)^{k+\frac{d-3}{2}}. \tag{A.6}$$

Here $\Gamma$ denotes the Gamma function, $\Gamma(z) = \int_0^\infty x^{z-1} e^{-x} \, dx$. Legendre polynomials are orthogonal on $[-1, 1]$ with respect to the measure with density $(1-t^2)^{(d-3)/2}$, which is the probability density function of the scalar product between to points on $\mathbb{S}^{d-1}$.

$$\int_{-1}^{+1} P_{k,d}(t) P_{k',d}(t) \left(1-t^2\right)^{\frac{d-3}{2}} dt = \frac{|\mathbb{S}^{d-1}|}{|\mathbb{S}^{d-2}|} \frac{\delta_{k,k'}}{\mathcal{N}_{k,s}}. \tag{A.7}$$

Here $|\mathbb{S}^{d-1}| = 2\pi^{\frac{d}{2}}/\Gamma(\frac{d}{2})$ denotes the surface area of the $d$-dimensional unit sphere ($|\mathbb{S}^0| = 2$ by definition).

To sum up, given $\boldsymbol{x}, \boldsymbol{y} \in \mathbb{S}^{d-1}$, functions of $\boldsymbol{x}$ or $\boldsymbol{y}$ can be expressed as a sum of projections on the orthonormal spherical harmonics, whereas functions of $\boldsymbol{x} \cdot \boldsymbol{y}$ can be expressed as a sum of projections on the Legendre polynomials. The relationship between the two expansions is elucidated in the Funk-Hecke formula [56, Thm. 2.22],

$$\int_{\mathbb{S}^{d-1}} f(\boldsymbol{x} \cdot \boldsymbol{y}) Y_{k,\ell}(\boldsymbol{y}) \, d\tau(\boldsymbol{y}) = Y_{k,\ell}(\boldsymbol{x}) \frac{|\mathbb{S}^{d-2}|}{|\mathbb{S}^{d-1}|} \int_{-1}^{+1} f(t) P_{k,d}(t) \left(1-t^2\right)^{\frac{d-3}{2}} dt := f_k Y_{k,\ell}(\boldsymbol{x}). \tag{A.8}$$

## A.1 Expansion of ReLU and combinations thereof

We can apply Eq. A.8 to have an expansion of neurons $\sigma\left(\boldsymbol{\theta}\cdot\boldsymbol{x}\right)$ in terms of spherical harmonics [2, Appendix D]. After defining

$$\varphi_k := \frac{|\mathbb{S}^{d-2}|}{|\mathbb{S}^{d-1}|} \int_{-1}^{+1} \sigma(t) P_{k,d}(t) \left(1-t^2\right)^{\frac{d-3}{2}} dt, \tag{A.9}$$

one has

$$\sigma\left(\boldsymbol{\theta}\cdot\boldsymbol{x}\right) = \sum_{k\geq 0} \mathcal{N}_{k,d}\varphi_k P_{k,d}\left(\boldsymbol{\theta}\cdot\boldsymbol{x}\right) = \sum_{k\geq 0} \varphi_k \sum_{\ell=1}^{\mathcal{N}_{k,d}} Y_{k,\ell}(\boldsymbol{\theta}) Y_{k,\ell}(\boldsymbol{x}). \tag{A.10}$$

For ReLU activations, in particular, $\sigma(t) = \max(0, t)$, thus

$$\varphi_k^{\text{ReLU}} = \frac{|\mathbb{S}^{d-2}|}{|\mathbb{S}^{d-1}|} \int_0^{+1} t P_{k,d}(t) \left(1-t^2\right)^{\frac{d-3}{2}} dt. \tag{A.11}$$

Notice that when $k$ is odd $P_{k,d}$ is an odd function of $t$, thus the integrand $t P_{k,d}(t)(1-t^2)^{\frac{d-3}{2}}$ is an even function of $t$. As a result the integral on the right-hand side of Eq. A.11 coincides with half the integral over the full domain $[-1, 1]$,

$$\int_0^{+1} t P_{k,d}(t) \left(1-t^2\right)^{\frac{d-3}{2}} dt = \frac{1}{2} \int_{-1}^{+1} t P_{k,d}(t) \left(1-t^2\right)^{\frac{d-3}{2}} dt = 0 \text{ for } k > 1, \tag{A.12}$$

because, due to Eq. A.7, $P_{k,d}$ is orthogonal to all polynomials with degree strictly lower than $k$. For even $k$ we can use Eq. A.6 and get [2] (see Eq. 3.3, main text)

$$\begin{aligned}
\int_0^{+1} t P_{k,d}(t) \left(1-t^2\right)^{\frac{d-3}{2}} dt &= \left(-\frac{1}{2}\right)^k \frac{\Gamma\left(\frac{d-1}{2}\right)}{\Gamma\left(k+\frac{d-1}{2}\right)} \int_0^1 t \frac{d^k}{dt^k} \left(1-t^2\right)^{k+\frac{d-3}{2}} dt \\
&= -\left(-\frac{1}{2}\right)^k \frac{\Gamma\left(\frac{d-1}{2}\right)}{\Gamma\left(k+\frac{d-1}{2}\right)} \frac{d^{k-2}}{dt^{k-2}} \left(1-t^2\right)^{k+\frac{d-3}{2}} \bigg|_{t=0}^{t=1} \\
&\Rightarrow \varphi_k^{\text{ReLU}} \sim k^{-\frac{d-1}{2}-\frac{3}{2}} \text{ for } k \gg 1 \text{ and even.}
\end{aligned} \tag{A.13}$$

Because all $\varphi_k^{\text{ReLU}}$ with $k > 1$ and odd vanish, even summing an infinite amount of neurons $\sigma\left(\boldsymbol{\theta}\cdot\boldsymbol{x}\right)$ with varying $\boldsymbol{\theta}$ does not allow to approximate any function on $\mathbb{S}^{d-1}$, but only those which have vanishing projections on all the spherical harmonics $Y_{k,\ell}$ with $k > 1$ and odd. This is why we set the odd coefficients of the target function spectrum to zero in Eq. 2.1.

## A.2 Dot-product kernels on the sphere

Also general dot-product kernels on the sphere admit an expansion such as Eq. A.10,

$$\mathcal{C}\left(\boldsymbol{x}\cdot\boldsymbol{y}\right) = \sum_{k\geq 0} \mathcal{N}_{k,d} c_k P_{k,d}\left(\boldsymbol{x}\cdot\boldsymbol{y}\right) = \sum_{k\geq 0} c_k \sum_{\ell=1}^{\mathcal{N}_{k,d}} Y_{k,\ell}(\boldsymbol{x}) Y_{k,\ell}(\boldsymbol{y}), \tag{A.14}$$

with

$$c_k = \frac{|\mathbb{S}^{d-2}|}{|\mathbb{S}^d|} \int_{-1}^1 \mathcal{C}(t) P_{k,d}(t) \left(1-t^2\right)^{\frac{d-3}{2}} dt. \tag{A.15}$$

The asymptotic decay of $c_k$ for large $k$ is controlled by the behaviour of $\mathcal{C}(t)$ near $t = \pm 1$, [58]. More precisely [58, Thm. 1], if $\mathcal{C}$ is infinitely differentiable in $(-1, 1)$ and has the following expansion around $\pm 1$,

$$\begin{cases} \mathcal{C}(t) = p_1(1-t) + c_1(1-t)^\nu + o\left((1-t)^\nu\right) \text{ near } t = +1; \\ \mathcal{C}(t) = p_{-1}(-1+t) + c_{-1}(-1+t)^\nu + o\left((-1+t)^\nu\right) \text{ near } t = -1, \end{cases} \tag{A.16}$$

where $p_{\pm 1}$ are polynomials and $\nu$ is not an integer, then

$$\begin{aligned}
k \text{ even: } & c_k \sim (c_1 + c_{-1}) k^{-2\nu-(d-1)}; \\
k \text{ odd: } & c_k \sim (c_1 - c_{-1}) k^{-2\nu-(d-1)},
\end{aligned} \tag{A.17}$$

The result above implies that that if $c_1 = c_{-1}$ ($c_1 = -c_{-1}$), then the eigenvalues with $k$ odd (even) decay faster than $k^{-2\nu-(d-2)}$. Moreover, if $\mathcal{C}$ is infinitely differentiable in $[-1, 1]$ then $c_k$ decays faster than any polynomial.

**NTK and RFK of one-hidden-layer ReLU networks**   Let $\mathbb{E}_{\boldsymbol{\theta}}$ denote expectation over a multivariate normal distribution with zero mean and unitary covariance matrix. For any $\boldsymbol{x}, \boldsymbol{y} \in \mathbb{S}^{d-1}$, the RFK of a one-hidden-layer ReLU network Eq. 2.3 with all parameters initialised as independent Gaussian random numbers with zero mean and unit variance reads

$$K^{\text{RFK}}(\boldsymbol{x} \cdot \boldsymbol{y}) = \mathbb{E}_{\boldsymbol{\theta}} \left[ \sigma(\boldsymbol{\theta} \cdot \boldsymbol{x}) \sigma(\boldsymbol{\theta} \cdot \boldsymbol{y}) \right]$$
$$= \frac{(\pi - \arccos{(t)})t + \sqrt{1 - t^2}}{2\pi}, \text{ with } t = \boldsymbol{x} \cdot \boldsymbol{y}. \tag{A.18}$$

The NTK of the same network reads, with $\sigma'$ denoting the derivative of ReLU or Heaviside function,

$$K^{\text{NTK}}(\boldsymbol{x} \cdot \boldsymbol{y}) = \mathbb{E}_{\boldsymbol{\theta}} \left[ \sigma(\boldsymbol{\theta} \cdot \boldsymbol{x}) \sigma(\boldsymbol{\theta} \cdot \boldsymbol{y}) \right] + (\boldsymbol{x} \cdot \boldsymbol{y}) \mathbb{E}_{\boldsymbol{\theta}} \left[ \sigma'(\boldsymbol{\theta} \cdot \boldsymbol{x}) \sigma'(\boldsymbol{\theta} \cdot \boldsymbol{y}) \right]$$
$$= \frac{2(\pi - \arccos{(t)})t + \sqrt{1 - t^2}}{2\pi}, \text{ with } t = \boldsymbol{x} \cdot \boldsymbol{y}. \tag{A.19}$$

As functions of a dot-product on the sphere, both NTK and RFK admit a decomposition in terms of spherical harmonics as Eq. A.15. For dot-product kernels, this expansion coincides with the Mercer's decomposition of the kernel [55], that is the coefficients of the expansion are the eigenvalues of the kernel. The asymptotic decay of the eigenvalues of such kernels $\varphi_k^{\text{NTK}}$ and $\varphi_k^{\text{RFK}}$ can be obtained by applying Eq. A.16 [58, Thm. 1]. Equivalently, one can notice that $K^{\text{RFK}}$ is proportional to the convolution on the sphere of ReLU with itself, therefore $\varphi_k^{\text{RFK}} = (\varphi_k^{\text{ReLU}})^2$. Similarly, the asymptotic decay of $\varphi_k^{\text{NTK}}$ can be related to that of the coefficients of $\sigma'$, derivative of ReLU: $\varphi_k(\sigma') \sim k\varphi(\sigma)$, thus $\varphi_k^{\text{NTK}} \sim k^2(\varphi_k^{\text{ReLU}})^2$. Both methods lead to Eq. 3.3 of the main text.

**Gaussian random fields and Eq. 2.2**   Consider a Gaussian random field $f^*$ on the sphere with covariance kernel $\mathcal{C}(\boldsymbol{x} \cdot \boldsymbol{y})$,

$$\mathbb{E}\left[f^*(\boldsymbol{x})\right] = 0, \quad \mathbb{E}\left[f^*(\boldsymbol{x})f^*(\boldsymbol{y})\right] = \mathcal{C}(\boldsymbol{x} \cdot \boldsymbol{y}), \quad \forall \boldsymbol{x}, \boldsymbol{y} \in \mathbb{S}^{d-1}. \tag{A.20}$$

$f^*$ can be equivalently specified via the statistics of the coefficients $f_{k,\ell}^*$,

$$\mathbb{E}\left[f_{k,\ell}^*\right] = 0, \quad \mathbb{E}\left[f_{k,\ell}^* f_{k',\ell'}^*\right] = c_k \delta_{k,k'} \delta_{\ell,\ell'}, \tag{A.21}$$

with $c_k$ denoting the eigenvalues of $\mathcal{C}$ in Eq. A.15. Notice that the eigenvalues are degenerate with respect to $\ell$ because the covariance kernel is a function $\boldsymbol{x} \cdot \boldsymbol{y}$: as a result, the random function $f^*$ is isotropic in law.

If $c_k$ decays as a power of $k$, then such power controls the weak differentiability (in the mean-squared sense) of the random field $f^*$. In fact, from Eq. A.4,

$$\left\| \Delta^{m/2} f^* \right\| = \sum_{k \geq 0} \sum_{\ell} (-k(k + d - 2))^m \left(f_{k,\ell}^*\right)^2. \tag{A.22}$$

Upon averaging over $f^*$ one gets

$$\mathbb{E}\left[\left\| \Delta^{m/2} f^* \right\|\right] = \sum_{k \geq 0} (-k(k + d - 2))^m \sum_{\ell} \mathbb{E}\left[\left(f_{k,\ell}^*\right)^2\right] = \sum_{k \geq 0} (-k(k + d - 2))^m \mathcal{N}_{k,d} c_k. \tag{A.23}$$

From Eq. A.16 [58, Thm. 1], if $\mathcal{C}(t) \sim (1-t)^{\nu_t}$ for $t \to 1$ and/or $\mathcal{C}(t) \sim (-1+t)^{\nu_t}$ for $t \to -1$, then $c_k \sim k^{-2\nu_t - (d-1)}$ for $k \gg 1$. In addition, for finite but arbitrary $d$, $(-k(k + d - 2))^m \sim k^{2m}$ and $\mathcal{N}_{k,s} \sim k^{d-2}$ (see Eq. A.1). Hence the summand in the right-hand side of Eq. A.23 is $\sim k^{2(m-\nu_t)-1}$, thus

$$\mathbb{E}\left[\left\| \Delta^{m/2} f^* \right\|\right] < \infty \quad \forall m < \nu_t. \tag{A.24}$$

Alternatively, one can think of $\nu_t$ as controlling the scaling of the difference $\delta f^*$ over inputs separated by a distance $\delta$. From Eq. A.20,

$$\mathbb{E}\left[|f^*(\boldsymbol{x}) - f^*(\boldsymbol{y})|^2\right] = 2\mathcal{C}(1) - 2\mathcal{C}(\boldsymbol{x} \cdot \boldsymbol{y}) = 2\mathcal{C}(1) + O((1 - \boldsymbol{x} \cdot \boldsymbol{y})^{\nu_t})$$
$$= 2\mathcal{C}(1) + O(|\boldsymbol{x} - \boldsymbol{y}|^{2\nu_t}) \tag{A.25}$$

## B   Uniqueness and Sparsity of the L1 minimizer

Recall that we want to find the $\gamma^*$ that solves

$$\gamma^* = \arg\min_\gamma \int_{\mathbb{S}^{d-1}} |d\gamma(\boldsymbol{\theta})| \quad \text{subject to} \quad \int_{\mathbb{S}^{d-1}} \sigma(\boldsymbol{\theta}\cdot\boldsymbol{x}_i)d\gamma(\boldsymbol{\theta})=f^*(\boldsymbol{x}_i) \quad \forall i = 1,\ldots,n. \quad \text{(B.1)}$$

In this appendix, we argue that the uniqueness of $\gamma^*$ which implies that it is atomic with at most $n$ atoms is a natural assumption. We start by discretizing the measure $\gamma$ into $H$ atoms, with $H$ arbitrarily large. Then the problem Eq. B.1 can be rewritten as

$$\boldsymbol{w}^* = \arg\min_{\boldsymbol{w}} \|\boldsymbol{w}\|_1, \quad \text{subject to} \quad \boldsymbol{\Phi w} = \boldsymbol{y}, \quad \text{(B.2)}$$

with $\boldsymbol{\Phi} \in \mathbb{R}^{H\times n}$, $\Phi_{h,i} = \sigma(\boldsymbol{\theta}_h\cdot\boldsymbol{x}_i)$ and $y_i = f^*(\boldsymbol{x}_i)$.

Given $\boldsymbol{w} \in \mathbb{R}^H$, let $\boldsymbol{u} = \max(\boldsymbol{w}, \boldsymbol{0}) \geq \boldsymbol{0}$ and $\boldsymbol{v} = -\max(-\boldsymbol{w}, \boldsymbol{0}) \geq \boldsymbol{0}$ so that $\boldsymbol{w} = \boldsymbol{u} - \boldsymbol{v}$. It is well-known (see e.g. [50]) that the minimization problem in (B.2) can be recast in terms of $\boldsymbol{u}$ and $\boldsymbol{v}$ into a linear programming problem. That is, $\boldsymbol{w}^* = \boldsymbol{u}^* - \boldsymbol{v}^*$ with

$$(\boldsymbol{u}^*, \boldsymbol{v}^*) = \arg\min_{\boldsymbol{u},\boldsymbol{v}} \boldsymbol{e}^T(\boldsymbol{u} + \boldsymbol{v}), \quad \text{subject to } \boldsymbol{\Phi u} - \boldsymbol{\Phi v} = \boldsymbol{y}, \quad \boldsymbol{u} \geq \boldsymbol{0}, \quad \boldsymbol{v} \geq \boldsymbol{0} \quad \text{(B.3)}$$

where $\boldsymbol{e} = [1, 1, \ldots, 1]^T$. Assuming that this problem is feasible (i.e. there is at least one solution to $\boldsymbol{\Phi u} - \boldsymbol{\Phi v} = \boldsymbol{y}$ such that $\boldsymbol{u} \geq \boldsymbol{0}$, $\boldsymbol{v} \geq \boldsymbol{0}$), it is known that it admits extremal solution, i.e. solutions such that at most $n$ entries of $(\boldsymbol{u}^*, \boldsymbol{v}^*)$ (and hence $\boldsymbol{w}^*$) are non-zero. The issue is whether such an extremal solution is unique. Assume that there are two, say $(\boldsymbol{u}_1^*, \boldsymbol{v}_1^*)$ and $(\boldsymbol{u}_2^*, \boldsymbol{v}_2^*)$. Then, by convexity,

$$(\boldsymbol{u}_t^*, \boldsymbol{v}_t^*) = (\boldsymbol{u}_1^*, \boldsymbol{v}_1^*)t + (\boldsymbol{u}_2^*, \boldsymbol{v}_2^*)(1 - t) \quad \text{(B.4)}$$

is also a minimizer of (B.3) for all $t \in [0, 1]$, with the same minimum value $\boldsymbol{u}_t^* + \boldsymbol{v}_t^* = \boldsymbol{u}_1^* + \boldsymbol{v}_1^* = \boldsymbol{u}_2^* + \boldsymbol{v}_2^*$. Generalizing this argument to the case of more than two extremal solutions, we conclude that all minimizers are global, with the same minimum value, and they live on the simplex where $\boldsymbol{e}^T(\boldsymbol{u} + \boldsymbol{v}) = \boldsymbol{e}^T(\boldsymbol{u}_1 + \boldsymbol{v}_1)$. Therefore, nonuniqueness requires that that this simplex has a nontrivial intersection with the feasible set where $\boldsymbol{\Phi u} - \boldsymbol{\Phi v} = \boldsymbol{y}$ with $\boldsymbol{u} \geq \boldsymbol{0}$, $\boldsymbol{v} \geq \boldsymbol{0}$. We argue that, generically, this will not be the case, i.e. the intersection will be trivial, and the extremal solution unique. In particular, since in our case we are in fact interested in the problem (B.1), we can always perturb slightly the discretization into $H$ atoms of $\gamma$ to guarantee that the extremal solution is unique. Since this is true no matter how large $H$ is, and any Radon measure can be approached to arbitrary precision using such discretization, we conclude that the minimizer of (B.1) should be unique as well, with at most $n$ atoms.

## C   Proof of Proposition 1

In this section, we provide the formal statement and proof of Proposition 1. Let us recall the general form of the predictor for both lazy and feature regimes in $d = 2$. From Eq. 3.6,

$$f^n(x) = \sum_{j=1}^n g_j\tilde{\varphi}(x - x_j) = \int \frac{dy}{2\pi} g^n(y)\tilde{\varphi}(x - y). \quad \text{(C.1)}$$

where $n$ is the number of training points for the lazy regime and the number of atoms for the feature regime and, for $x \in (-\pi, \pi]$,

$$\tilde{\varphi}(x) = \begin{cases} \max\{0, \cos(x)\} & \text{(feature regime)}, \\ \dfrac{2(\pi - |x|)\cos(x) + \sin(|x|)}{2\pi} & \text{(lazy regime, NTK)}, \\ \dfrac{(\pi - |x|)\cos(x) + \sin(|x|)}{2\pi} & \text{(lazy regime, RFK)}. \end{cases} \quad \text{(C.2)}$$

All these functions $\tilde{\varphi}$ have jump discontinuities on some derivative: the first for feature and NTK, the third for RFK. If the $l$-th derivative has jump discontinuities, the $l + 1$-th only exists in a distributional sense and it can be generically written as a sum of a regular function and a sequence of Dirac masses

located at the discontinuities. With $m$ denoting the number of such discontinuities and $\{x_j\}_j$ their locations, $f^{(l)}$ denoting the $l$-th derivative of $f$, for some $c_j \in \mathbb{R}$,

$$f^{(l+1)}(x) = f_r^{(l+1)}(x) + \sum_{j=1}^{m} c_j \delta(x - x_j), \tag{C.3}$$

where $f_r$ denotes the *regular* part of $f$.

**Proposition 2.** *Consider a random target function $f^*$ satisfying Eq. 2.1 and the predictor $f^n$ obtained by training a one-hidden-layer ReLU network on $n$ samples $(x_i, f^*(x_i))$ in the feature or in the lazy regime (Eq. C.1). Then, with $\widehat{f}(k)$ denoting the Fourier transform of $f(x)$, one has*

$$\lim_{|k| \to \infty} \lim_{n \to \infty} \frac{\widehat{(f^n)_r''}(k)}{\widehat{f^*}(k)} = c, \tag{C.4}$$

*where $c$ is a constant (different for every regime). This result implies that as $n \to \infty$, $(f^n)''(x)$ converges to a function having finite second moment, i.e.*

$$\begin{aligned}
\lim_{n \to \infty} \mathbb{E}_{f^*} [(f^n)_r''(x)]^2 &= \lim_{n \to \infty} \mathbb{E}_{f^*} \left[ \int dx \, ((f^n)_r'')^2 (x) \right] \\
&= \lim_{n \to \infty} \mathbb{E}_{f^*} \left[ \sum_k \widehat{(f^n)_r''}^2 (k) \right] = const. < \infty,
\end{aligned} \tag{C.5}$$

*using the fact that $\mathbb{E}_{f^*}[(f^n)_r''(x)]^2$ does not depend on $x$ and $\mathbb{E}_{f^*}[\sum_k \widehat{(f^*)}^2 (k)] = const.$*

*Proof:* Because our target functions are random fields that are in $L_2$ with probability one, and the RKHS of our kernels are dense in that space, we know that the test error vanishes as $n \to \infty$ [59]. As a result

$$f^*(x) = \lim_{n \to \infty} f^n(x) = \lim_{n \to \infty} \int \frac{dy}{2\pi} g^n(y) \tilde{\varphi}(x - y). \tag{C.6}$$

Consider first the feature regime and the NTK lazy regime. In both cases $\tilde{\varphi}$ has two jump discontinuities in the first derivative, located at $x = 0, \pi$ for the NTK and at $x = \pm \pi/2$, therefore we can write the second derivative as the sum of a regular function and two Dirac masses,

$$\begin{aligned}
(\tilde{\varphi}^{\text{FEATURE}})'' &= -\max\{0, \cos(x)\} + \delta(x - \pi/2) + \delta(x + \pi/2), \\
(\tilde{\varphi}^{\text{NTK}})'' &= \frac{-2(\pi - |x|)\cos(x) + 3\sin(|x|)}{2\pi} - \frac{1}{2\pi}\delta(x) + \frac{1}{2\pi}\delta(x - \pi).
\end{aligned} \tag{C.7}$$

As a result, the second derivative of the predictor can be written as the sum of a regular part $(f^n)_r''$ and a sequence of $2n$ Dirac masses. After subtracting the Dirac masses, both sides of Eq. C.1 can be differentiated twice and yield

$$(f^n)_r''(x) = \int \frac{dy}{2\pi} g^n(y) \tilde{\varphi}_r''(x - y). \tag{C.8}$$

Hence in the Fourier representation we have

$$\widehat{(f^n)_r''}(k) = \widehat{g^n}(k)(-k^2 \widehat{\tilde{\varphi}}_r(k)) \tag{C.9}$$

where we defined

$$\widehat{\tilde{\varphi}}(k) = \int_{-\pi}^{\pi} \frac{dx}{\sqrt{2\pi}} e^{ikx} \tilde{\varphi}(x), \qquad \widehat{\tilde{\varphi}_r}(k) = \int_{-\pi}^{\pi} \frac{dx}{\sqrt{2\pi}} e^{ikx} \tilde{\varphi}_r(x). \tag{C.10}$$

and used $\widehat{\tilde{\varphi}_r''}(k) = -k^2 \widehat{\tilde{\varphi}_r}(k)$. By universal approximation we have

$$\widehat{f^*}(k) = \int_{-\pi}^{\pi} \frac{dx}{\sqrt{2\pi}} e^{ikx} f^*(x) = \lim_{n \to \infty} \widehat{g^n}(k)\widehat{\tilde{\varphi}}(k) \qquad \Rightarrow \qquad \lim_{n \to \infty} \widehat{g^n}(k) = \frac{\widehat{f^*}(k)}{\widehat{\tilde{\varphi}}(k)}. \tag{C.11}$$

As a result by combining Eq. C.9 and Eq. C.11 we deduce

$$\lim_{n \to \infty} \widehat{(f^n)_r''}(k) = -\frac{k^2 \widehat{\tilde{\varphi}}_r(k)}{\widehat{\tilde{\varphi}}(k)} \widehat{f^*}(k). \tag{C.12}$$

To complete the proof using this result it remains to estimate the scaling of $\widehat{\tilde{\varphi}}_r(k)$ and $\widehat{\tilde{\varphi}}(k)$ in the large $|k|$ limit.

For the feature regime, a direct calculation shows that $\tilde{\varphi}''_r = -\tilde{\varphi}$, implying that $\widehat{\tilde{\varphi}}_r(k) = -\widehat{\tilde{\varphi}}(k)$. This proves that Eq. C.4 is satisfied with $c = -1$.

For the NTK lazy regime $\tilde{\varphi}''_r$ and $-\tilde{\varphi}$ are different but they have similar singular expansions near $x = 0$ and $\pi$. Therefore their Fourier coefficients display the same asymptotic decay. More specifically, with $t = \cos(x)$ (or $x = \arccos(t)$), so that $\tilde{\varphi}(x) = \varphi(t)$, one has

$$\begin{cases} \varphi^{\mathrm{NTK}}(t) = t - \dfrac{1}{\sqrt{2}\pi}(1-t)^{1/2} + O\left((1-t)^{3/2}\right) \text{ near } t = +1; \\[2mm] \varphi^{\mathrm{NTK}}(t) = -\dfrac{1}{\sqrt{2}\pi}(-1+t)^{1/2} + O\left((-1+t)^{3/2}\right) \text{ near } t = -1, \end{cases} \tag{C.13}$$

and

$$\begin{cases} (\varphi^{\mathrm{NTK}})''_r(t) = -t + \dfrac{5}{\sqrt{2}\pi}(1-t)^{1/2} + O\left((1-t)^{3/2}\right) \text{ near } t = +1; \\[2mm] (\varphi^{\mathrm{NTK}})''_r(t) = +\dfrac{5}{\sqrt{2}\pi}(-1+t)^{1/2} + O\left((-1+t)^{3/2}\right) \text{ near } t = -1. \end{cases} \tag{C.14}$$

Therefore, due to Eq. A.17, Eq. C.4 is satisfied with $c = -5$. The same procedure can be applied to the RFK lazy regime, with the exception that it is the fourth derivative of $\tilde{\varphi}^{\mathrm{RFK}}$ which can be written as a regular part plus Dirac masses, but one can still obtain the Fourier coefficients of the second derivative's regular part by dividing those of the fourth derivative's regular part by $k^2$.

## D Asymptotics of generalization in $d = 2$

In this section we compute the decay of generalization error $\bar{\epsilon}$ with the number of samples $n$ in the following 2-dimensional setting:

$$f^n(x) = \sum_{j=1}^n g_j \tilde{\varphi}(x - x_j), \tag{D.1}$$

where the $x_j$'s are the training points (like in the NTK case) and $\varphi$ has a single discontinuity on the first derivative in 0.

Let us order the training points clockwise on the ring, such that $x_1 = 0$ and $x_{i+1} > x_i$ for all $i = 1, \ldots, n$, with $x_{n+1} := 2\pi$. On each of the $x_i$ the predictor coincides with the target,

$$f^n(x_i) = f^*(x_i) \quad \forall \, i = 1, \ldots, n. \tag{D.2}$$

For large enough $n$, the difference $x_{i+1} - x_i$ is small enough such that, within $(x_i, x_{i+1})$, $f^n(x)$ can be replaced with its Taylor series expansion up to the second order. In practice, the predictors appear like the cable of a suspension bridge with the pillars located on the training points. In particular, we can consider an expansion around $x_i^+ := x_i + \epsilon$ for any $\epsilon > 0$ and then let $\epsilon \to 0$ from above:

$$f^n(x) = f^n(x_i^+) + (x - x_i^+)f^{n\prime}(x_i^+) + \frac{(x - x_i^+)^2}{2}(f^n)''(x_i^+) + \mathcal{O}\left((x - x_i^+)^3\right). \tag{D.3}$$

By differentiability of $f^n$ in $(x_i, x_{i+1})$ the second derivative can be computed at any point inside $(x_i, x_{i+1})$ without changing the order of approximation in Eq. D.3, in particular we can replace $(f^n)''(x_i^+)$ with $c_i$, the mean curvature of $f^n$ in $(x_i, x_{i+1})$. Moreover, as $\epsilon \to 0$, $f^n(x_i^+) \to f^*(x_i)$ and $f^n(x_{i+1}^-) \to f^*(x_{i+1})$. By introducing the limiting slope $m_i^+ := \lim_{x \to 0+} f^{n\prime}(x_i + x)$, we can write

$$f^n(x) = f^*(x_i) + (x - x_i)m_i^+ + \frac{(x - x_i)^2}{2}c_i + O\left((x - x_i^+)^3\right) \tag{D.4}$$

Computing Eq. D.4 at $x = x_{i+1}$ yields a closed form for the limiting slope $m_i^+$ as a function of the mean curvature $c_i$, the interval length $\delta_i := (x_{i+1} - x_i)$ and $\Delta f_i := f^*(x_{i+1}) - f^*(x_i)$. Specifically,

$$m_i^+ = \frac{\Delta f_i}{\delta_i} - \frac{\delta_i}{2}c_i. \tag{D.5}$$

The generalization error can then be split into contributions from all the intervals. If $\nu_t > 2$, A Taylor expansion leads to:

$$\epsilon(n) = \int_0^{2\pi} \frac{dx}{2\pi} \left(f^n(x) - f^*(x)\right)^2$$

$$= \sum_{i=1}^n \int_{x_i}^{(x_{i+1})} \frac{dx}{2\pi} \left[(x - x_i)\left(m_i^+ - (f^*)'(x_i)\right) + \frac{(x - x_i)^2}{2}\left(c_i - (f^*)''(x_i)\right) + o\left((x - x_i^+)^2\right)\right]^2$$

$$= \sum_{i=1}^n \int_0^{\delta_i} \frac{d\delta}{2\pi} \left[\delta\left(m_i^+ - (f^*)'(x_i)\right) + \frac{\delta^2}{2}\left(c_i - (f^*)''(x_i)\right) + o\left(\delta^2\right)\right]^2$$

$$= \sum_{i=1}^n \frac{1}{2\pi} \left[\frac{\delta_i^3}{3}\left(m_i^+ - (f^*)'(x_i)\right)^2 + \frac{\delta_i^5}{20}\left(c_i - (f^*)''(x_i)\right)^2\right.$$

$$\left. + \frac{\delta_i^4}{4}\left(m_i^+ - (f^*)'(x_i)\right)\left(c_i - (f^*)''(x_i)\right) + o(\delta_i^5)\right].$$

$$(D.6)$$

In addition, as $\Delta f_i = (f^*)'(x_i)\delta_i + (f^*)''(x_i)\delta_i^2/2 + O(\delta_i^3)$,

$$m_i^+ - (f^*)'(x_i) = \frac{\delta_i}{2}\left((f^*)''(x_i) - c_i\right) + o(\delta_i)^2, \tag{D.7}$$

thus

$$\epsilon(n) = \frac{1}{2\pi} \sum_{i=1}^n \left[\frac{\delta_i^5}{120}\left(c_i - (f^*)''(x_i)\right)^2 + o(\delta_i^5)\right]. \tag{D.8}$$

implying:

$$\overline{\epsilon}(n) = \frac{n^{-4}\left(n^{-1}\sum_{i=1}^n (n\delta_i)^5\right)}{240\pi} \lim_{n\to\infty} \int \mathbb{E}_{f^*}\left[\left((f^n)''(x) - (f^*)''(x)\right)^2\right] dx + o(n^{-4}) \sim \frac{1}{n^4} \tag{D.9}$$

where we used that *(i)* the integral converges to some finite value, due to proposition 2. From [App. C](#), this integral can be estimated as $\sum_k \mathbb{E}_{f^*}\left[\left(cf^*(k) - k^2 f^*(k)\right)^2\right]$, that indeed converges for $\nu_t > 2$. *(ii)* $\left(n^{-1}\sum_{i=1}^n (n\delta_i)^5\right)$ has a deterministic limit for large $n$. It is clear for the lazy regime since the distance between adjacent singularities $\delta_i$ follows an exponential distribution of mean $\sim \frac{1}{n}$. We expect this result to be also true for the feature regime in our set-up. Indeed, in the limit $n \to \infty$, the predictor approaches a parabola between singular points, which generically cannot fit more than three random points. There must thus be a singularity at least every two data-points with a probability approaching unity as $n \to \infty$, which implies that $\left(n^{-1}\sum_{i=1}^n (n\delta_i)^5\right)$ converges to a constant for large $n$.

Finally, for $\nu_t < 2$, the same decomposition in intervals applies, but a Taylor expansion to second order does not hold. The error is then dominated by the fluctuations of $f^*$ on the scale of the intervals, as indicated in the main text.

## E  Asymptotic of generalization via the spectral bias ansatz

According to the spectral bias ansatz, the first $n$ modes of the predictor $f_{k,\ell}^n$ coincide with the modes of the target function $f_{k,\ell}^*$. Therefore, the asymptotic scaling of the error with $n$ is entirely controlled by the remaining modes,

$$\epsilon(n) \sim \sum_{k \geq k_c} \sum_{\ell=1}^{\mathcal{N}_{k,d}} \left(f_{k,\ell}^n - f_{k,\ell}^*\right)^2 \quad \text{with} \quad \sum_{k \leq k_c} \mathcal{N}_{k,d} \sim n. \tag{E.1}$$

Since $\mathcal{N}_{k,d} \sim k^{d-2}$ for $k \gg 1$, one has that, for large $n$, $k_c \sim n^{\frac{1}{d-1}}$. After averaging the error over target functions we get

$$\overline{\epsilon}(n) \sim \sum_{k \geq k_c} \sum_{\ell=1}^{\mathcal{N}_{k,d}} \left\{\mathbb{E}_{f^*}\left[\left(f_{k,\ell}^n\right)^2\right] + \mathbb{E}_{f^*}\left[\left(f_{k,\ell}^*\right)^2\right] - 2\mathbb{E}_{f^*}\left[\left(f_{k,\ell}^n f_{k,\ell}^*\right)\right]\right\}. \tag{E.2}$$

Let us recall that, with the predictor having the general form in Eq. 3.2, then

$$f_{k,\ell}^n = g_{k,\ell}^n \varphi_k \quad \text{with} \quad g_{k,\ell}^n = \sum_{j=1}^n g_j Y_{k,\ell}(\boldsymbol{y}_j), \tag{E.3}$$

where the $\boldsymbol{y}_j$'s denote the training points for the lazy regime and the neuron features for the feature regime. For $k \ll k_c$, where $f_{k,\ell}^n = f_{k,\ell}^*$, $g_{k,\ell}^n = f_{k,\ell}^*/\varphi_k$. For $k \gg k_c$, due to the highly oscillating nature of $Y_{k,\ell}$, the factors $Y_{k,\ell}(\boldsymbol{y}_j)$ are essentially decorrelated random numbers with zero mean and finite variance, since the values of $(Y_{k,\ell}(\boldsymbol{y}_j))^2$ are limited by the addition theorem Eq. A.5. Let us denote the variance with $\sigma_Y$. By the central limit theorem, $g_{k,\ell}^n$ converges to a Gaussian random variable with zero mean and finite variance $\sigma_Y^2 \sum_{j=1}^n g_j^2$. As a result,

$$\begin{aligned}
\overline{\epsilon}(n) &\sim \sum_{k \geq k_c} \sum_{\ell=1}^{\mathcal{N}_{k,d}} \left\{ \left( \sum_{j=1}^n g_j^2 \right) \varphi_k^2 + \mathbb{E}_{f^*}\left[ \left(f_{k,\ell}^*\right)^2 \right] \right\} \\
&= \left( \sum_{j=1}^n g_j^2 \right) \sum_{k \geq k_c} \mathcal{N}_{k,d}\varphi_k^2 + \sum_{k \geq k_c} \mathcal{N}_{k,d}c_k,
\end{aligned} \tag{E.4}$$

where we have used the definition of $f^*$ (Eq. 2.1) to set the expectation of $(f_{k,\ell}^*)^2$ to $c_k$.

**Large $\nu_t$ case** When $f^*$ is smooth enough the error is controlled by the predictor term proportional to $\sum_{j=1}^n g_j^2$. More specifically, if

$$\sum_{k \geq 0} \sum_{\ell=1}^{\mathcal{N}_{k,d}} \frac{c_k}{\varphi_k^2} < +\infty, \tag{E.5}$$

then the function $g^n(\boldsymbol{x})$ converges to the square-summable function $g^*(\boldsymbol{x})$ such that $f^*(\boldsymbol{x}) = \int g^*(\boldsymbol{y})\varphi(\boldsymbol{x} \cdot \boldsymbol{y})\,d\tau(\boldsymbol{y})$. With $c_k \sim k^{-2\nu_t-(d-1)}$ and $\mathcal{N}_{k,d} \sim k^{d-2}$, in the lazy regime $\varphi_k \sim k^{-(d-1)-2\nu}$ Eq. E.5 is satisfied when $2\nu_t > 2(d-1) + 4\nu$ ($\nu = 1/2$ for the NTK and $3/2$ for the RFK). In the feature regime $\varphi_k \sim k^{-(d-1)/2-3/2}$, Eq. E.5 is satisfied when $2\nu_t > (d-1) + 3$. If $g^n(\boldsymbol{x})$ converges to a square-summable function, then

$$\sum_{j=1}^n g_j^2 = \frac{1}{n} \int g^n(\boldsymbol{x})^2\,d\tau(\boldsymbol{x}) + o(n^{-1}) = \frac{1}{n} \sum_{k \geq 0} \mathcal{N}_{k,d}\frac{c_k}{\varphi_k^2} + o(n^{-1}), \tag{E.6}$$

which is proportional to $n^{-1}$. In addition, since $\mathcal{N}_{k,d} \sim k^{d-2}$ and $k_c \sim n^{\frac{1}{d-1}}$, one has

$$n^{-1} \sum_{k \geq k_c} \mathcal{N}_{k,d}\varphi_k \sim \begin{cases} n^{-1} k^{d-1} k^{-2(d-1)-4\nu}\big|_{k=n^{\frac{1}{d-1}}} \sim n^{-2-\frac{4\nu}{d-1}} \text{ (Lazy)}, \\ n^{-1} k^{d-1} k^{-(d-1)-3}\big|_{k=n^{\frac{1}{d-1}}} \sim n^{-1-\frac{3}{d-1}} \text{ (Feature)}, \end{cases} \tag{E.7}$$

and

$$\sum_{k \geq k_c} \mathcal{N}_{k,d}c_k \sim k^{d-1} k^{-2\nu_t-(d-1)}\big|_{k=n^{\frac{1}{d-1}}} \sim n^{-\frac{2\nu_t}{d-1}}. \tag{E.8}$$

Hence, if $\nu_t$ is large enough so that Eq. E.5 is satisfied, the asymptotic decay of the error is given by Eq. E.7.

**Small $\nu_t$ case** If Eq. E.7 does not hold then $g^n(\boldsymbol{x})$ is not square-summable in the limit $n \to \infty$. However, for large but finite $n$ only the modes up to the $k_c$-th are correctly reconstructed, therefore

$$\sum_{j=1}^n g_j^2 \sim \frac{1}{n} \sum_{k \leq k_c} \mathcal{N}_{k,d}\frac{c_k}{\varphi_k^2} \sim \begin{cases} n^{-1} k^{-2\nu_t} k^{2(d-1)+4\nu}\big|_{k=n^{\frac{1}{d-1}}} \sim n^{-\frac{2\nu_t}{d-1}} n^{1+\frac{4\nu}{d-1}} \text{ (Lazy)}, \\ n^{-1} k^{-2\nu_t} k^{(d-1)+3}\big|_{k=n^{\frac{1}{d-1}}} \sim n^{-\frac{2\nu_t}{d-1}} n^{\frac{3}{d-1}} \text{ (Feature)}, \end{cases} \tag{E.9}$$

Both for feature and lazy, multiplying the term above by $\sum_{k \geq k_c} \mathcal{N}_{k,d}\varphi_k$ from Eq. E.7 yields $\sim n^{-2\nu_t/(d-1)}$. This is also the scaling of the target function term Eq. E.8, implying that for small $\nu_t$ one has

$$\overline{\epsilon}(n) \sim n^{-\frac{2\nu_t}{d-1}} \tag{E.10}$$

both in the feature and in the lazy regimes.

# F  Spectral bias via the replica calculation

Due to the equivalence with kernel methods, the asymptotic decay of the test error in the lazy regime can be computed with the formalism of [45], which also provides a non-rigorous justification for the spectral bias ansatz. By ranking the eigenvalues from the biggest to the smallest, such that $\varphi_\rho$ denotes the $\rho$-th eigenvalue and denoting with $c_\rho$ the variance of the projections of the target onto the $\rho$-th eigenfunction, one has

$$\epsilon(n) = \sum_\rho \epsilon_\rho(n), \quad \epsilon_\rho(n) = \frac{\kappa(n)^2}{(\varphi_\rho + \kappa(n))^2} c_\rho, \quad \kappa(n) = \frac{1}{n}\sum_\rho \frac{\varphi_\rho \kappa(n)}{\varphi_\rho + \kappa(n)}. \tag{F.1}$$

It is convenient to introduce the eigenvalue density,

$$\mathcal{D}(\varphi) := \sum_{k \geq 0}\sum_{l=1}^{\mathcal{N}_{k,d}} \delta(\varphi - \varphi_k) = \sum_{k\geq 0} \mathcal{N}_{k,d}\delta(\varphi - \varphi_k) \sim \int_0^\infty k^{d-2}\delta(\varphi - k^{-(d-1)-2\nu})\, \text{for } k \gg 1. \tag{F.2}$$

After changing variables in the delta function, one finds

$$\mathcal{D}(\varphi) \sim \varphi^{-\frac{2(d-1)+2\nu}{(d-1)+2\nu}} \text{ for } \varphi \ll 1. \tag{F.3}$$

This can be used for inferring the asymptotics of $\kappa(n)$,

$$\begin{aligned}
\kappa(n) = \frac{1}{n}\sum_\rho \frac{\varphi_\rho \kappa(n)}{\varphi_\rho + \kappa(n)} &\sim \frac{1}{n}\int d\varphi\, \mathcal{D}(\varphi)\frac{\varphi\kappa(n)}{\varphi + \kappa(n)} \\
&\sim \frac{1}{n}\int_0^{\kappa(n)} d\varphi\, \mathcal{D}(\varphi)\varphi + \frac{\kappa(n)}{n}\int_{\kappa(n)}^{\varphi_0} d\varphi\, \mathcal{D}(\varphi) \\
&\sim \frac{1}{n}\kappa(n)^{1-\frac{(d-1)}{(d-1)+2\nu}} \Rightarrow \kappa(n) \sim n^{-1-\frac{2\nu}{d-1}}.
\end{aligned} \tag{F.4}$$

Once the scaling of $\kappa(n)$ has been determined, the modal contributions to the error can be split according to whether $\varphi_\rho \ll \kappa(n)$ or $\varphi_\rho \gg \kappa(n)$. The scaling of $\varphi_\rho$ with the rank $\rho$ is determined self-consistently,

$$\rho \sim \int_{\varphi_\rho}^{\varphi_1} d\varphi\, \mathcal{D}(\varphi) \sim \varphi_\rho^{-\frac{d-1}{(d-1)+2\nu}} \Rightarrow \varphi_\rho \sim \rho^{-1-\frac{2\nu}{d-1}} \Rightarrow \varphi_\rho \gg (\ll)\kappa(n) \Leftrightarrow \rho \ll (\gg)n. \tag{F.5}$$

Therefore

$$\epsilon(n) \sim \kappa(n)^2 \sum_{\rho \ll n}\frac{c_\rho}{\varphi_\rho^2} + \sum_{\rho \gg n} c_\rho. \tag{F.6}$$

Notice that $\kappa(n)^2$ scales as $n^{-1}\sum_{k \geq k_c}\mathcal{N}_{k,s}\varphi_k$ in Eq. E.7, whereas $\sum_{\rho \ll n} c_\rho/\varphi_\rho^2$ corresponds to $n\sum_j g_j^2$ in Eq. E.9, so that the first term on the right-hand side of Eq. F.6 matches that of Eq. E.4. The same matching is found for the second term on the right-hand side of Eq. F.6, so that the replica calculation justifies the spectral bias ansatz.

# G  Training wide neural networks: does gradient descent (GD) find the minimal-norm solution?

In the main text we provided predictions for the asymptotics of the test error of the minimal norm solution that fits all the training data. Does the prediction hold when solution of Eq. 2.5 and Eq. 2.13 is approximately found by GD? More specifically, is the solution found by GD the minimal-norm one?

**Feature Learning**  We answer these questions by performing full-batch gradient descent in two settings (further details about the trainings are provided in the code repository, `experiments.md` file),

1. **Min-L1.** Here we update weights and features of Eq. 2.3, with $\xi = 0$, by following the negative gradient of

$$\mathcal{L}_{\text{Min-L1}} = \frac{1}{2n} \sum_{i=1}^{n} \left( f^*(\boldsymbol{x}_i) - f(\boldsymbol{x}_i) \right)^2 + \frac{\lambda}{H} \sum_{h=1}^{H} |w_h|, \tag{G.1}$$

   with $\lambda \to 0^+$. The weights $w_h$ are initialized to zero and the features are initialized uniformly and constrained to be on the unit sphere.

2. **$\alpha$-trick.** Following [8], here we minimize

$$\mathcal{L}_{\alpha\text{-trick}} = \frac{1}{2n\alpha} \sum_{i=1}^{n} \left( f^*(\boldsymbol{x}_i) - \alpha f(\boldsymbol{x}_i) \right)^2, \tag{G.2}$$

   with $\alpha \to 0$. This trick allows to be far from the lazy regime by forcing the weights to evolve to $\mathcal{O}(1/\alpha)$, when fitting a target of order 1.

In both cases, the solution found by GD is sparse, in the sense that is supported on a finite number of neurons – in other words, the measure $\gamma(\boldsymbol{\theta})$ becomes atomic, satisfying Assumption 1. Furthermore, we find that

1. For **Min-L1**, the generalization error prediction holds (Fig. 4 and Fig. G.1) as the the minimal norm solution if effectively recovered, see Fig. G.2. Such clean results in terms of features position are difficult to achieve for large $n$ because the training dynamics becomes very slow and reaching convergence becomes computationally infeasible. Still, we observe the test error to plateau and reach its infinite-time limit much earlier than the parameters, which allows for the scaling predictions to hold.

2. **$\alpha$-trick**, however, does not recover the minimal-norm solution, Fig. G.2. Still, the solution found is of the type (2.7) as it is sparse and supported on a number of atoms that scales linearly with $n$, Fig. G.3, left. For this reason, we find that our predictions for the generalization error hold also in this case, see Fig. G.3, right.

**Lazy Learning**   In this case, the correspondence between the solution found by gradient descent and the minimal-norm one is well established [9]. Therefore, numerical experiments are performed here via kernel regression and the analytical NTK Eq. A.19: given a dataset $\{\boldsymbol{x}_i, y_i = f^*(\boldsymbol{x}_i)\}_{i=1}^{n}$, we define the gram matrix $\mathbf{K} \in \mathbb{R}^{n \times n}$ with elements $\mathbf{K}_{ij} = K(\boldsymbol{x}_i, \boldsymbol{x}_j)$ and the vector of target labels $\boldsymbol{y} = [y_1, y_2, \ldots, y_n]$. The $q_i$'s in Eq. 2.9 can be easily recovered by solving the linear system

$$\boldsymbol{y} = \frac{1}{n} \mathbf{K} \boldsymbol{q}. \tag{G.3}$$

**Experiments**   Numerical experiments are run with PyTorch on GPUs NVIDIA V100 (university internal cluster). Details for reproducing experiments are provided in the code repository, experiments.md file. Individual trainings are run in 1 minute to 1 hour of wall time. We estimate a total of a thousand hours of computing time for running the preliminary and actual experiments present in this work.

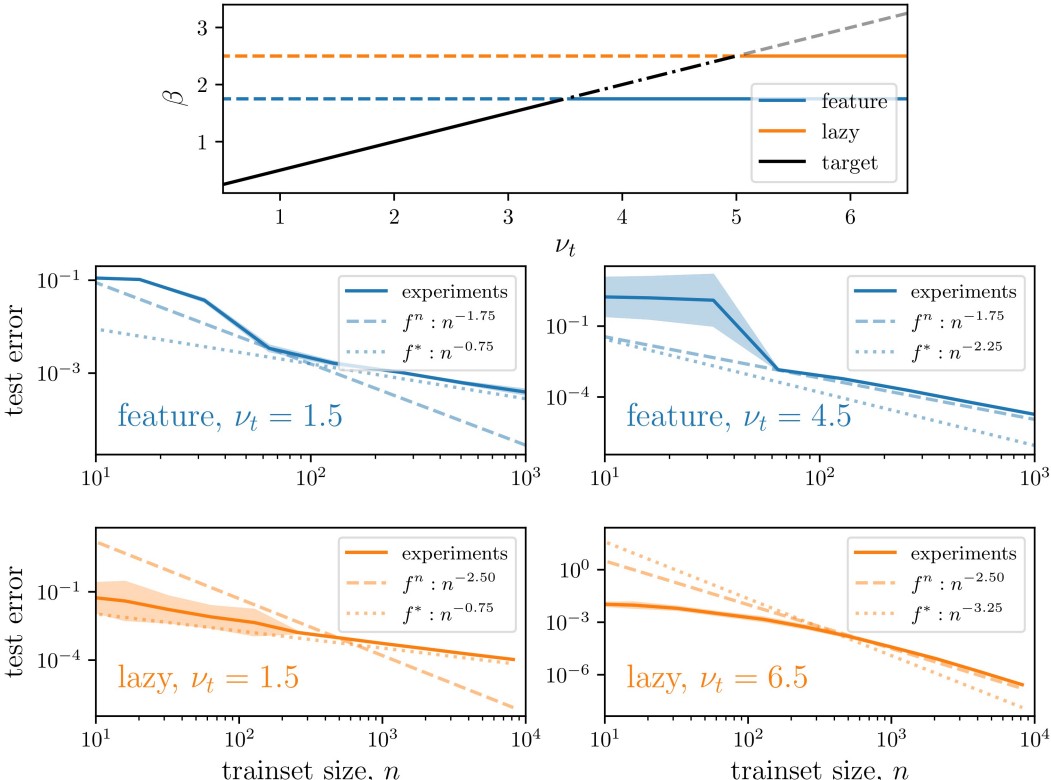

Figure G.1: **Gen. error decay vs. target smoothness and training regime.** Here, data-points are sampled uniformly from the spherical surface in $d = 5$ and the target function is an infinite-width FCN with activation function $\sigma(\cdot) = |\cdot|^{\nu_t - 1/2}$, corresponding to a Gaussian random process of smoothness $\nu_t$. 1ˢᵗrow: gen. error decay exponent as a function of the target smoothness $\nu_t$. The three curves correspond to the target contribution to the generalization error (black) and the predictor contribution in either feature (blue) or lazy (orange) regime. Full lines highlight the dominating contributions to the gen. error. 2ⁿᵈrow: agreement between predictions and experiments in the feature regime for a non-smooth (left) and smooth (right) target. In the first case, the error is dominated by the target $f^*$, in the second by the predictor $f^n$ – predicted exponents $\beta$ are indicated in the legends. 3ʳᵈrow: analogous of the previous row for the lazy regime.

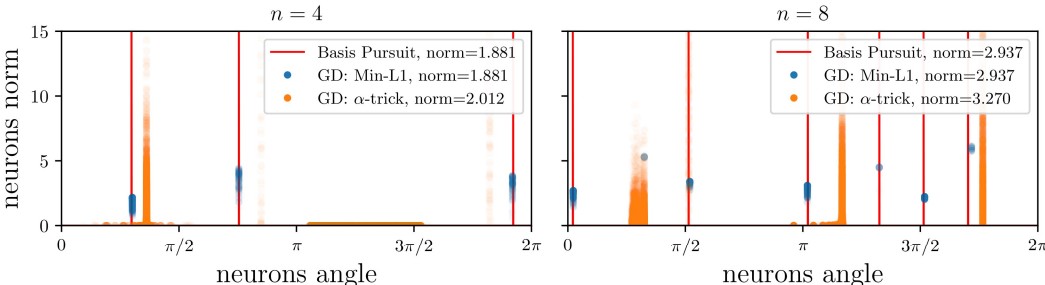

Figure G.2: **Comparing solutions.** Solutions to the spherically symmetric task in $d = 2$ for $n = 4$ (left) and $n = 8$ (right) training points. In red the minimal norm solution (Eq. 2.5) as found by Basis Pursuit [50]. Solutions found by GD in the Min-L1 and $\alpha$-trick setting are respectively shown in blue and orange. Dots correspond to single neurons in the network. The $x$-axis reports their angular position while the $y$-axis reports their norm: $|w_h| \|\boldsymbol{\theta}_h\|_2$. The total norm of the solutions, $\frac{\alpha}{H} \sum_{h=1}^{H} |w_h| \|\boldsymbol{\theta}_h\|_2$, is indicated in the legend.

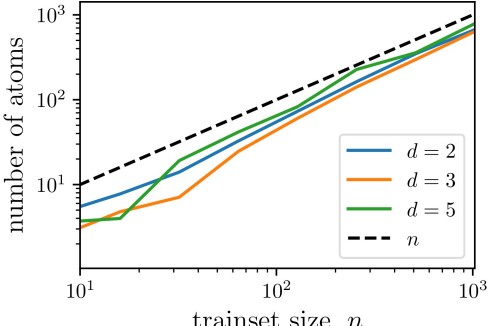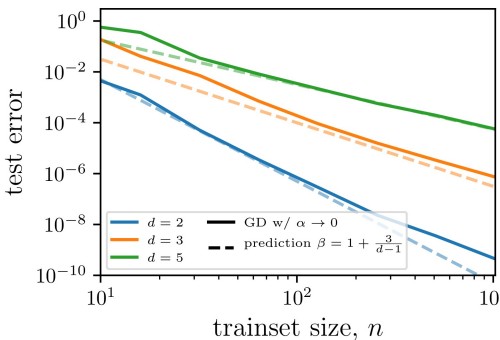

Figure G.3: **Solution found by the $\alpha$-trick.** We consider here the case of approximating the constant target function on $\mathbb{S}^{d-1}$ with an FCN. Training is performed starting from small initialization through the $\alpha$-trick. Left: Number of atoms $n_A$ as a function of the number of training points $n$. Neurons that are active on the same subset of the training set are grouped together and we consider each group a distinct atom for the counting. Right: Generalization error in the same setting (full), together with the theoretical predictions (dashed). Different colors correspond to different input dimensions. The case of $d = 2$ and large $n$ suffers from the same finite time effects discussed in Fig. 4. Results are averaged over 10 different initializations of the networks and datasets.

## H  Sensitivity of the predictor to transformations other than diffeomorphisms

This section reports experiments to integrate the discussion of section 5. In particular, we: *(i)* show that the lazy regime predictor is less sensitive to image translations than the feature regime one (as is the case for deformations, from Fig. 6); *(ii)* provide evidence of the positive effects of learning features in image classifications, namely becoming invariant to pixels at the border of images which are unrelated to the task.

To prove the above points we consider, as in Fig. 6, the relative sensitivity of the predictors of lazy and feature regime with respect to global translations for point *(i)* and corruption of the boundary pixels for point *(ii)*. The relative sensitivity to translations is obtained from Eq. 5.1 after replacing the transformation $\tau$ with a one-pixel translation of the image in a random direction. For the relative sensitivity to boundary corruption, the transformation consists in adding zero-mean and unit-variance Gaussian numbers to the boundary pixels. Both relative sensitivities are plotted in Fig. H.1, with translations on the left and boundary pixels corruption on the right.

In section 5 we then argue that differences in performance between the two training regimes can be explained by gaps in sensitivities with respect to input transformations that do not change the label. For *(i)*, the gap is similar to the one observed for diffeomorphisms (Fig. 6). Still, the space of translations has negligible size with respect to input space, hence we expect the diffeomorphisms to have a more prominent effect. In case *(ii)*, the feature regime is less sensitive with respect to irrelevant pixels corruption and this would give it an advantage over the lazy regime. The fact that the performance difference is in favor of the lazy regime instead, means that these transformations only play a minor role.

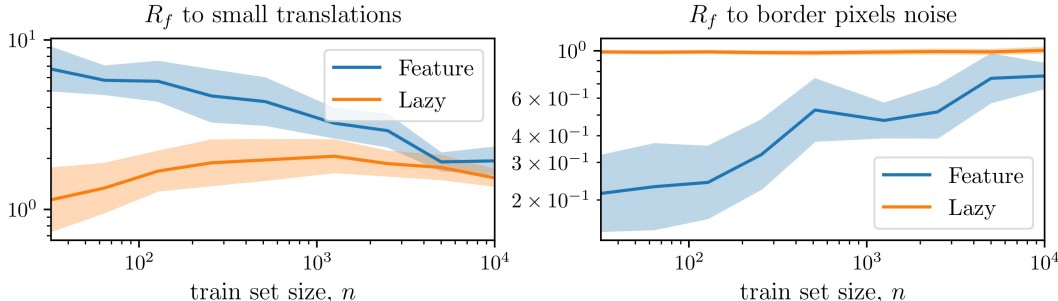

Figure H.1: **Sensitivity to input transformations vs number of training points.** Relative sensitivity of the predictor to (left) random 1-pixel translations and (right) white noise added at the boundary of the input images, in the two regimes, for varying number of training points $n$ and when training on FashionMNIST. Smaller values correspond to a smoother predictor, on average. Results are computed using the same predictors as in Fig. 1. Left: For small translations, the behavior is the same compared to applying diffeomorphisms. Right: The lazy regime does not distinguish between noise added at the boundary or on the whole image ($R_f = 1$), while the feature regime gets more insensitive to the former.

# I Maximum-entropy model of diffeomorphisms

We briefly review here the maximum-entropy model of diffeomorphisms as introduced in [49].

An image can be thought of as a function $x(s)$ describing intensity in position $s = (u, v) \in [0, 1]^2$, where $u$ and $v$ are the horizontal and vertical (pixel) coordinates. Denote $\tau x$ the image deformed by $\tau$, i.e. $[\tau x](s) = x(s - \tau(s))$. [49] propose an ensemble of diffeomorphisms $\tau(s) = (\tau_u, \tau_v)$ with i.i.d. $\tau_u$ and $\tau_v$ defined as

$$\tau_u = \sum_{i,j \in \mathbb{N}^+} C_{ij} \sin(i\pi u) \sin(j\pi v) \tag{I.1}$$

where the $C_{ij}$'s are Gaussian variables of zero mean and variance $T/(i^2 + j^2)$ and $T$ is a parameter controlling the deformation magnitude. Once $\tau$ is generated, pixels are displaced to random positions. See Fig. 5b for an example of such transformation.