# OpenReview forum: "Learning sparse features can lead to overfitting in neural networks"
_NeurIPS.cc/2022/Conference — NeurIPS 2022 Accept_

### Official Review · Reviewer_Rp2y · 2022-07-12

**Rating:** 6
**Confidence:** 3
**Soundness:** 3 good
**Presentation:** 4 excellent
**Contribution:** 3 good

**Summary:**

The papers provide insights into why learning features in a certain setting are not robust (may overfit) compared to the lazy training regime. They show that learning features results in sparse features, and they argue that when the target function is smooth along certain input dimensions, this is detrimental. Hence, the non-learning features method generalizes better.

**Questions:**

1. See my comment above on wording for generalization/robustness. The statements in this regard are confusing and should be rephrased. The paper mainly talks about robustness, not the generalization to train distribution.

2. Provide supporting argument/applications where the target function is smooth (e.g., for line 48).

3. What is the role of overparameterization?

Minor. On line 39, the author claims that feature learning outperforms lazy training in tasks such as approximating a function. Please demonstrate this similar to Figure 1.

**Strengths And Weaknesses:**

I enjoyed reading the paper. The presentation is clear and well organized. It includes proper experiments with good visualization and comparisons. See my comments below for improvement.

- Confusion of several terminologies. The paper uses generalization terminology. However, it covers the robustness of their methods against specific deformations without providing literature on robustness. Moreover, the discussed deformations may be seen as robustness or out-of-distribution generalization. Hence, it is not completely proper to phrase that learning-feature networks with sparse representations "overfit".

- The paper argues that sparse features are not good in fully connected, but they are good in CNN (there is sparsity in CNN and such networks contain robust sparse features). Why this is the case? The results are not convincing of the fact that sparsity is the leading factor in the lack of robustness.

- The sparsity of the features is related to the neural collapse. Please include a discussion related to this and generalization. Please specify at which training size n, the networks become overparameterized.

- This paper talks about sparsity. However, it is missing literature on the generalization of sparse coding [1,2].


[1] Mehta, Nishant, and Alexander Gray. "Sparsity-based generalization bounds for predictive sparse coding." International Conference on Machine Learning. PMLR, 2013.

[2] Sulam, Jeremias, Ramchandran Muthukumar, and Raman Arora. "Adversarial robustness of supervised sparse coding." Advances in Neural Information Processing Systems 33 (2020): 2110-2121.

Minor: In Figure 5, the label should be "train set".

---

> ### Author Response · Authors · 2022-08-02
> **Reply to reviewer Rp2y**
>
> We thank the referee for their comments and review. We reply to the single comments below.
>
> **Confusion on generalization and robustness terminology.** We believe that the reviewer misunderstood our aim here. What we seek to explain is a paradox on the traditional notion of generalization, and compare lazy and feature regimes. We show that lazy can beat feature learning (in the sense of achieving better generalization) if the task requires a continuous distribution of neurons to be represented. We argue that this is the case if the task is stable to smooth deformations. To test for the stability of predictors, we consider deformations of very mild amplitude (i.e. such that each pixel in the image is displaced by 1, on average). Yet, we never use the deformed images for training or for measuring generalization.
>
> **Learning features in FCNs vs CNNs.** As stressed in the previous point, we believe robustness is not the good terminology here. Also, our work is not about CNNs. We simply report (and do not seek to explain) the fact that, in deep CNNs, learning features allows the network to build a low-dimensional representation of data, which is presumably beneficial for generalization (see also discussion with reviewer ZBQb). More generally, the advantages of learning features in deep networks are not well-understood, and we believe this is one of the main open questions of the field. In this work, instead, we seek to explain the drawbacks of learning features. We start from an observation already present in the literature, then establish that the same phenomenon is observed in a simple data model, where it can be understood in analytical terms. We support our arguments with numerical tests.
>
> **On the relation to neural collapse.** please refer to the third point of our answer to reviewer ZBQb. All our theoretical results refer to the scrict overparametrized limit where the number of hidden neurons is infinitely large whereas the number of training points $n$ is large but finite.
>
> **On sparse coding.** We will follow the reviewer suggestion and cite the relevant literature in the final version of the manuscript.
>
> **On smoothness of the target.** For image data, we expect the target function to be smooth e.g. for exact translations, small rotations, changes in brightness or contrast. To answer the reviewers we tested that our approach also holds for pure translations.
>
> **On overparametrization.** We study the infinite-width limit for two reasons: first, overparametrized neural networks are observed to work best in practice; secondly, the theoretical analysis of generalization presented in the paper is only possible in this limit. In practice, to make sure we are measuring properties of that limit, we repeat our simulations both for two widths W and 2*W, and check that measurements of quantities of interest (e.g. exponent $\beta$) did not change.
>
> **On the setting where feature beats lazy.** More precisely, the claim we make is that “for certain tasks, including approximating a function which depends only on a subset or a linear combination of the input variables,” feature outperforms lazy. This is a well-established result [10, 15-18]. See in particular [18, Figure 3] for a plot analogous to Figure 1, in the case of a target function $f^*(x_1, x_2, …, x_d) = f^*(x_1)$.

---

> > ### Comment · Reviewer_Rp2y · 2022-08-04
> > **After Initial Rebuttal**
> >
> > I thank the authors for addressing my comments. I still see that the paper needs a clarification on robustness and generalization. To reduce this confusion, I suggest to include the provided explanation by authors in the manuscript. In addition, please include the smoothness discussion, and also the clarifications on where feature beats lazy. Given the authors' response to all reviews, I recommend acceptance of this paper.

---

> > > ### Author Response · Authors · 2022-08-07
> > > **Thanks**
> > >
> > > We thank the reviewer for their feedback. We will include the highlighted discussions and clarifications in the final manuscript, where one more content page is allowed.

---

### Official Review · Reviewer_ZBQb · 2022-07-12

**Rating:** 7
**Confidence:** 4
**Soundness:** 3 good
**Presentation:** 3 good
**Contribution:** 3 good

**Summary:**

This paper studies the dynamics of how a neural network learns from data. Authors focus on the claim that neural networks learn sparse representation of data, and demonstrate that when the underlying phenomenon is not sparse, encouraging the model to learn sparse features will lead to bad generalization.


**Questions:**

Please see the comments above.

**Limitations:**

Overall, the arguments put forward in the paper do not naturally generalize to real world applications of neural networks on images and such. This is a limitation that can be acknowledged in a clearer way.

**Strengths And Weaknesses:**

Strengths:
Paper is well written and its arguments are clear. Its message seems intuitive, even trivial, but it seems to put into perspective some of the ideas that are present in the literature. Overall, it seems to me that the research community is better off having this paper in the literature, but I would like to see it significantly improved before publication.


Weaknesses:
The main argument seems trivial to me: if the underlying phenomenon is not sparse, one should not try to learn sparse features, or try to impose sparsity into the learning process.

The notion of feature learning vs lazy learning seems fluid and ambiguous. It is not clear why we should categorize learning into these two types, and if we do this categorization, where the exact line would be that separates these two categories of feature learning vs lazy learning.

Paper does not consider the argument that models learn a "low-dimensional feature space" from the original data, rather it focuses on the argument that models learn "sparse features". These two arguments are not equivalent, but they are related, and authors can at least acknowledge those studies on the literature. For example, see:

Cohen, U., Chung, S., Lee, D.D. and Sompolinsky, H., 2020. Separability and geometry of object manifolds in deep neural networks. Nature communications, 11(1), pp.1-13.

Zarka, J., Guth, F. and Mallat, S., 2021, May. Separation and Concentration in Deep Networks. In ICLR 2021-9th International Conference on Learning Representations.

Bengio, Y., Courville, A. and Vincent, P., 2013. Representation learning: A review and new perspectives. IEEE transactions on pattern analysis and machine intelligence, 35(8), pp.1798-1828.

Many of the arguments in the literature that suggest models learn a sparse representation, imply that models learn a low dimensional feature space. So, sparsity may not be the property of that low-dimensional feature space, rather, it could be the property of original data.

For example, consider a dataset that has linearly dependent features and therby, its data matrix is rank deficient. By dropping the zero singular values, one can project that data into a lower dimensional space without losing any information. The data and its projection may not be sparse at all.

Paper does not consider the studies that impose sparsity on the parameters of the model. For example, this paper shows that convolutional layers can be learned using feed-forward networks and imposing sparsity on the parameters of the model:

Neyshabur, B., 2020. Towards learning convolutions from scratch. Advances in Neural Information Processing Systems, 33, pp.8078-8088.

When authors talk about high dimensions, especially in introduction and in the closing, it is not clear what they actually mean. How many dimensions are considered high dimensions, and how many dimensions would be low? Some of these arguments seem hand wavy and ambiguous.

FCN is not defined in the paper the first time it is introduced.

---

> ### Author Response · Authors · 2022-08-02
> **Reply to reviewer ZBQb**
>
> We thank the referee for their careful review and for the useful comments. Below we provide answers to the specific questions.
>
> **On the simplicity of the main argument.** The observation that the lazy regime can beat feature learning for image datasets was known, but not understood. The originality of the paper is to provide an explanation for this previous observation, based on the following ideas: (i) images class varies little along smooth deformations of the image; (ii) due to that, tasks like image classification require a continuous distribution of neurons to be represented; (iii) thus, requiring sparsity diminishes performance. This last idea (which is only one step in the argument) is simple, which we view as a strength rather than a weakness. Moreover, a quantitative treatment of the effect of sparsity on learning curves (generalization error vs. number of training examples) was absent in the literature.
>
> **On the notion of feature vs lazy.** There is indeed a continuum of possibilities that can be explored by varying the initialization scale (which is indeed a continuous parameter). Generalization performance displays a smooth crossover between the two extreme regimes considered, when the weights evolution, in norm, is of the same order of the weights magnitude at initialization, i.e. $\|w(T) - w(0)\| / \|w(0)\| \sim O(1)$,  as documented in the paper below. Here we focus on the two extreme regimes as they both admit a simple formulation that allows us to predict generalization performance. Let us stress that even if we only consider extreme regimes, our predictions allow us to understand a phenomenon that is observed also away from such extreme limits, see M. Geiger, S. Spigler, A. Jacot, and M. Wyart. Disentangling feature and lazy training in deep neural networks. Journal of Statistical Mechanics: Theory and Experiment, 2020(11):113301.
>
> **On the advantages of learning sparse features and related literature.** One of the possible advantages of Feature learning (FL) is indeed the possibility of learning low-dimensional representations of the data. This property is arguably crucial for performance, especially in deep networks. To our knowledge, though, it is mostly understood in the case of one-hidden-layer neural networks and we cite relevant literature in this context [10-18]. Understanding what are the advantages of FL in deep networks, in our perspective, is still one of the main open problems of deep learning theory, and we are not pretending to tackle that with this work. Our central goal here is to seek to explain some observations indicating that sometimes FL can be detrimental. That said, we are happy to better review the literature about the positive aspects of feature learning and the advantages of learning sparse representations, when the task allows for it. We will include a section on this literature in the final manuscript, including the neural collapse phenomenon, manifold learning, learning locality from scratch by imposing sparsity.
>
> **High dimension and the curse of dimensionality.**  By “curse of dimensionality” we refer to the fact that, generically, the exponent $\beta$ should scale as an inverse power of the dimension, i.e. $\beta = 1/d$. In practice, though, $\beta \simeq 1/10$ is observed for image classification tasks where $d = 10^3$ to $10^5$ (e.g. J. Hestness, S. Narang, N. Ardalani, G> F. Diamos, H. Jun, H. Kianinejad, Md. M. A. Patwary, Y. Yang, and Y. Zhou. Deep learning scaling is predictable, empirically. CoRR, abs/1712.00409, 2017). The high-dimensional examples we are interested in are then the cases in which the curse is unexpectedly beaten, i.e. the measured $\beta \gg 1/d$.

---

> > ### Comment · Reviewer_ZBQb · 2022-08-03
> > **Clarifications**
> >
> > Thanks - your clarifications and improvements in the paper address most of my concerns. It would be helpful to reflect these clarifications in the paper as well. For example, the point about the contributions of the paper is clear enough in rebuttal but perhaps not as clear in the paper. Similarly, for the notion of feature vs lazy, and for high dimensionality, it would be helpful to include your arguments in the paper.
> >
> > I wonder if authors have considered the projection of data in the last hidden layer of trained networks (which has the least number of dimensions in a network architecture). For example, figure 4(a) depicts the distribution of CIFAR-10 dataset in the feature space projected to a 2d space via PCA. Do authors have insights about the projection of CIFAR-10 dataset in the last hidden layer of a trained model? How many principal components (i.e., singular values) were significant? Would the discussion in Section 5 apply to the feature space of the last hidden layer?

---

> > > ### Author Response · Authors · 2022-08-07
> > > **Reply to reviewer ZBQb**
> > >
> > > We will include additional paragraphs to the final manuscript where one more content page is allowed. In particular, we will add a quantitative discussion on dimensions in the introduction, a paragraph to section 'our contribution' to summarize the main argument, and another paragraph motivating our choice of extreme limits as representative of feature and lazy regimes.
> > >
> > > With respect to the question on projections on the last hidden layer: at least for one-hidden-layer networks, the features collapse shown in Fig. 4 suggests that the last hidden layer representation is indeed low-dimensional. We have performed additional experiments on a one-hidden-layer FCN trained on CIFAR-10 that suggest that the number of principal components that have significant singular value is small compared to the width of the final layer, i.e. order 10 vs order 1000. In practice, the first few principal components are much more informative of the classes for architectures with better performance (e.g. CNN vs FCN), as reported by the neural collapse literature, Papyan, Vardan, X. Y. Han, and David L. Donoho. "Prevalence of neural collapse during the terminal phase of deep learning training." Proceedings of the National Academy of Sciences 117.40 (2020).
> > > Regarding the discussion in Section 5, the relative stability of the last hidden layer is very similar to the output one, as reported in L. Petrini, A. Favero, M. Geiger, and M. Wyart. "Relative stability toward diffeomorphisms indicates performance in deep nets". In Advances in Neural Information Processing Systems 2021.

---

### Official Review · Reviewer_vPhN · 2022-07-13

**Rating:** 7
**Confidence:** 4
**Soundness:** 3 good
**Presentation:** 4 excellent
**Contribution:** 4 excellent

**Summary:**

The authors consider the asymptotic behavior of the generalization error in two
different regimes of idealized (infinite-width) neural network training: the
"feature regime" where the network has a small initialization and
regularization on the parameters, and the "lazy regime" where the
initialization scale is larger and the network behaves like a kernel ridge
regression method in the limit. The setting they study is regression from iid
samples of an isotropic gaussian random field on the sphere, with data and
hidden layer weights uniformly distributed on the sphere. The results show that
when the smoothness of the target function is sufficiently high, the asymptotic
generalization error of the learned predictor in the kernel regime has a
superior scaling exponent to that of the feature regime predictor: the
intuition given is that the feature regime induces a sparse learned predictor,
which is inferior for fitting very smooth functions. They connect the notion of
fitting highly smooth predictors to invariance to small diffeomorphisms in
image classification, and present numerical experiments that suggest that a
similar performance gap is observable there.



**Questions:**

In connection with the discussion of small diffeomorphism stability, it would
be interesting if the authors can point out connections or implications to
other works that study deep networks in kernel regimes for computing with
low-dimensional data models, such as manifold classification [1] or RF kernel
regression on subspheres [2], relative to learning such data in the feature
regime.

I do not quite understand why Assumption 1 (the uniqueness part) is necessary
here. There seems to be a significant amount of generic structure on the
problem (2.5), given that the covariates are uniformly random and the target
function is also random; in light of the heuristic argument in appendix B, why
is it not possible to exploit this structure to argue that (2.5) has a unique
solution?  This is a standard process in similar finite-dimensional problems
arising in e.g. sparse approximation with $\ell^1$ regularization and random
independent measurements -- I am just curious why similar ideas can't be
instantiated here to remove this assumption.

What can be said (qualitatively) about the behavior of predictors that are
smooth enough for (3.4a, b) to show an advantage for lazy regime training?
Evidently we need $\nu_t$ to grow with $d$ -- does this become equivalent to
the predictor being in the kernel's RKHS? (e.g. Sobolev embedding theorems and
RKHS equivalence results?)

[1] https://openreview.net/forum?id=O-6Pm_d_Q-

[2] http://arxiv.org/abs/2006.13409


**Limitations:**

Yes.

**Strengths And Weaknesses:**

## Strengths

- The paper presents an interesting finding that offers an alternate viewpoint
  to one of the predominant ones in the DL theory literature -- that the
  two-layer "feature learning" training regime is inherently superior to the
  kernel regime -- and supports the settings where this does not hold by
  arguing that invariance to nuisance transformations in image classification
  might be such a setting. This will be an interesting addition to this
  conversation in general.
- The paper is clearly written. Although technical, it provides ample
  well-chosen references to background/related works that allow technical
  aspects of the problem to be understood. Intuitions are explained so clearly
  that the conclusions of the paper end up feeling natural after some thought.
- The experiments presented are thought-provoking and make a connection to
  practical structured data in a way that many in the deep learning theory
  literature do not.

## Weaknesses

- Proofs are somewhat terse. I could not follow the proof of Proposition 2
  around eqn C.8 very well -- is there any need to justify the interchanging of
  limits/integrals here? The calculation in section D appears to apply to
  $\varphi$ with a single discontinuity, rather than to the functions
  associated to the NTK or the ReLU. Some clarification on whether the
  calculations in Appendices D and E are being claimed as rigorous might be
  helpful -- this word is used in the main paper at points, and the intuitive
  discussions make sense, but some of the arguments appear heuristic (perhaps
  this is just because the details are often left to the reader).
- It might make sense to examine some of the hypotheses around diffeomorphism
  stability in a more 'toy dataset' setting relative to just testing on
  benchmark datasets (since the issue of varying the train set size is somewhat
  strange here, given non-iid-ness, and the performance separations in Figure 5
  do not always support the hypothesis). For example, what about creating a
  synthetic dataset of small diffeomorphisms (or even moderate-dimensional
  parametric deformations, which will be easier to sample and compute) of two a
  fixed templates embedded into some background, and trying to classify?
  Additionally, here a metric that truly captures diffeomorphism stability
  could be used rather than the relative one of eqn (5.1).

## Minor Issues

- Hyperref links are broken in the pdf. I also cannot access the github
  repository link in line 69. Although there may be other fixes, I always enjoy
  using the `xr` package in Latex to avoid this.
- I did not follow how the Lazy regime described in lines 123-135 relates to the
  training of an actual neural network for fitting the data (it is not clear to
  me how the "Assume that..." at the start of the paragraph is linked to an
  actual neural network training; the definition (2.3) also has a
  nonstandard-seeming initial "lazy regime" network, which is identically
  zero).  I only mention this because of the contrast with the description of
  the feature regime above, where eqn (2.6) makes the link fairly
  well-motivated. Since the experiments for "lazy regime" are eventually just
  simulating the associated KRR method, perhaps the description could be
  amended accordingly, and/or a reference could be added that describes the
  connection to neural network fitting with this initialization scheme.
- I was not familiar with the precise result alluded to in lines 110-112; would
  be slightly helpful to add a more precise pointer e.g.  [31, Section 5]
  perhaps.

---

> ### Author Response · Authors · 2022-08-02
> **Reply to reviewer vPhN**
>
> We thank the reviewer for their careful and useful review. Some of the comments have been implemented in a revised version of the paper with changes highlighted in blue.
>
> ## Weaknesses
>
> **Terse proofs.** We have uploaded a revised version of the paper with a simpler version of proposition 2 and a more detailed proof in Appendix. The calculations in Appendix D are indeed rigorous in the simplified case where $\varphi$ has a single discontinuity (as we specify in the main paper, line 156 and lines 180-181); but we expect the same result to hold for NTK and ReLU as argued in Appendix (lines 639-644) and systematically checked in numerical experiments.
>
> **toy dataset.** Regarding Fig. 5, we agree that deformation stability alone is not enough to explain generalization in real data, and other (competing) effects are at play (see lines 267-268).  We find the synthetic setting proposition very interesting. If one chooses two fixed templates that represent classes and give inter-class variability by deformations, though, the following issue arises: FCNs could exploit spurious features of the data for classification without actually learning deformation invariance, e.g. pixels in the bulk of the templates could always be black for one class and white for the other if the applied deformations are small. We tried to build such a dataset and ended up having such issues.
>
> ## Minor issues
>
> 1. (broken links) Hyperref links broke when we split the pdf into main text and appendix for the submission. We apologize for the inconvenience and provide here the link to the anonymized [GitHub repository](https://anonymous.4open.science/r/regressionsphere-DB74/README.md);
> 2. (definition of lazy regime) We address the reviewer's concerns and suggestions in the revised version;
> 3. (precise pointer to ref. 31) The reviewer is correct, we are referring to section 5 of [31], more precisely theorem 10. We will specify this in the revised manuscript.
>
> ## Questions
>
> **Small deformation stability and low-dimensional data models.** We thank the reviewer for pointing out these references. [1] determines how deep networks disentangle data that live on separate smooth manifolds embedded in a higher-dimensional space. What we probe here, instead, is how the predictor changes when moving on the manifold itself (along diffeo), as evidence supports that stability along such transformations is related to generalization (small change when moving on the manifold correlates with good generalization properties). The setting of [2] is also different from ours, as it considers a target function that depends on a linear combination of the input coordinates. For this case, it is well known that learning features leads to better generalization, but this intuition cannot be extended to problems where the low-dimensional features depend nonlinearly on the input coordinates (as occurs for smooth transformations).
>
> **On the necessity of assumption 1.** We agree with the reviewer’s comment, however, the argument presented in appendix B requires a discretization of the Radon measure. Although this discretization can be arbitrarily fine, it is not clear to us how to extend the argument to the continuum limit.
>
> **Eq. 3.4 vs RKHS condition.** The condition for the target to belong to the RKHS is $\nu_t > nu + (d-1)/2$ and the crossover occurs at $\nu_t = 2\nu + (d-1)$ in Eq. 3.4a ($\nu_t$ twice as large as the minimum for belonging to the RKHS) and $\nu_t = 3/2 + (d-1)/2$ in Eq.3.4b. Therefore, in the feature regime, the crossover occurs when the target belongs to the RKHS of the random feature kernel (for which $\nu=3/2$).

---

> > ### Comment · Reviewer_vPhN · 2022-08-04
> > **thanks**
> >
> > Dear authors,
> >
> > Thank you for your response to my review, and your hard work in updating the submission.
> >
> > - I am satisfied with the clarifications to the introduction of the NTK regime predictor on page 4.
> > - The revisions to the proofs look good to me.
> > - Thanks for responding to the speculative suggestion about synthetic data. What about embedding the deformed templates into random (e.g. uniform/gaussian noise) backgrounds, attempting to capture a setting where the network cannot afford to memorize the background -- does this introduce a different issue? Perhaps this also has an interesting connection to the relative stability $R_f$ (or, since I have not thought carefully about it, doesn't work because of some kind of confounding effect that is captured by this $R_f$ definition). This setting makes me think of the experiments of [1] -- I wonder if you could build off of this kind of experimental apparatus in your setting.
> >
> > [1] S. Karp, E. Winston, Y. Li, and A. Singh, “Local Signal Adaptivity: Provable Feature Learning in Neural Networks Beyond Kernels,” in Advances in Neural Information Processing Systems, 2021, vol. 34, pp. 24883–24897.

---

> > > ### Author Response · Authors · 2022-08-07
> > > **On the synthetic data model**
> > >
> > > We thank the reviewer for the stimulating propositions.
> > >
> > > We think that background noise would not solve the issue as two cases emerge: either the noise magnitude is small and we go back to the zero-noise case, or it is large enough to prevent the network from learning thus just affecting accuracy. Some preliminary experiments suggest this to be the case but cannot exclude that a non-trivial intermediate noise regime exists for which the reviewer's intuition is verified. Establishing that would require a more thorough experimentation with such dataset.
> > > The main point of the task proposed in [1] is denoising, while the task we aim at building is one in which deformation invariance is needed for learning. For this reason, we do not see a straightforward connection between the two settings.

---

### Official Review · Reviewer_mho4 · 2022-07-14

**Rating:** 6
**Confidence:** 2
**Soundness:** 3 good
**Presentation:** 3 good
**Contribution:** 3 good

**Summary:**

The paper tackles an important question of the impact of feature learning on generalization. While it is a common wisdom that feature learning is beneficial, experimentally it is not the case for fully-connected networks on image datasets.

The authors show that when learning a random isotropic Gaussian process on a sphere with a wide, single-hidden-layer ReLU network, models trained in the feature and lazy regimes will exhibit different scaling (with the dataset size) of the generalization MSE error. The scaling depends on the smoothness of the target function, and for sufficiently smooth functions the lazy regime will actually scale better.

The authors verify their theory with experiments, and provide some discussion and further experiments connecting these results to the observed phenomena in practice (that lazy fully-connected networks perform better on image datasets than feature learning fully-connected networks).


**Questions:**

- Could you provide a companion plot to Figure 1 plotting classification error instead of the MSE?

- On line 111 you say that the feature regime is equivalent to L2 regularization. Does your result then imply that on image datasets as considered in Figure 1, fully-connected networks shouldn't use weight decay? If so, how does this reconcile with the common wisdom that weight decay tends to improve generalization (see e.g. top row of Figure 1 in https://arxiv.org/pdf/2007.15801.pdf)?

- Could you elaborate on your reasoning on lines 245-246, i.e. why the low dimension of the transformation makes it negligible? Even though translations are 2D, aren't they still not learnable by a few neurons, hence could be a good space of transformations to test your theory? Same for rotations (which are 1D).

	* Relatedly, it would also be nice to contrast Figure 5 with equivalent plots for deformations that are learnable with a few neurons (perhaps brightness, corruption to a few fixed pixels, etc).

- Is there an explanation for why in Figure 5 $R_f$ is usually $> 1$, i.e. networks are more sensitive to image diffeomorphisms than to random noise?

- Why is $R_f$ chosen to be the ratio of sensitivities, as opposed to just the numerator? Intuitively, in connection to generalization, I would expect the absolute (vs relative) stability to matter. Relatedly, in Figure 1 error curves for CIFAR-10 diverge for large $n$, but stability curves, in contrast, converge for large $n$ (with lazy even becoming more sensitive than feature all the way to the right), which casts doubt on stability explaining the generalization gap here.

- Line 292: which best architectures don't use pooling?

- Do you have thoughts on connecting more explicitly $\nu_t$ to some metric of the dataset like CIFAR-10? Perhaps, fitting a Gaussian process with smoothness $\nu_t$, picking/optimizing $\nu_t$ for max likelihood, then using it to predict whether lazy or feature will do better?

**Limitations:**

All good.

**Strengths And Weaknesses:**

The paper presents a specific and novel result on the importance of feature learning in deep learning, which is also well-backed by experiments, hence I recommend acceptance.

The evidence and discussion on transferring these results to real-world datasets appear promising but preliminary, which is why my recommendation is less strong.

## Pros:

- Interesting and important topic.
- Clear and novel result, well-backed by experiments.
- Well-written.

## Cons:

- Considered construction is rather specific, and it will require more future work to understand how relevant these results are to explaining the actual generalization gaps on real datasets observed in practice (e.g. in Figure 1). The authors propose an interesting hypothesis in section 5, but I find it inconclusive (see also some questions below). While more connections to practical settings would make this a much stronger paper, I find the presented results already significant enough for publication.

## Minor

- Definition of stability (Eq. 5.1) is somewhat counter-intutivie, since high stability actually implies high sensitivity (ergo low stability) to $\tau$. Also, superscript $n$ can be omitted to declutter.

- All links, including the code repository, are not clickable.

- I would use more visually-distinct line styles in Figure G.1 (e.g. `:` vs `-.`).

---

> ### Author Response · Authors · 2022-08-02
> **Reply to reviewer moh4**
>
> We thank the referee for their comments and reply to each of them below.
>
> **On the defintion of stability.** We will use the word sensitivity in place of stability and omit the superscript as suggested.
>
> **Broken links.** Hyperref links broke when we split the pdf into main and appendix for submission. We apologize for the incovenience and provide here a link to the anonymized [GitHub repository](https://anonymous.4open.science/r/regressionsphere-DB74/README.md)
>
> **Line styles in figure G.1.** We will replace dash-dotted lines with dotted lines as suggested.
>
> **Figure 1.** The y-axis does report the classification error (i.e. fraction of misclassified test samples).
>
> **Role of weight decay**. The equivalence between L1 and L2 regularization is only valid for one-hidden-layer fully-connected networks, while [1] considers 3-layers neural networks, which can change the role of weight decay – in deep networks, weight decay may help discover locality from data, for example. Moreover, [1] considers SGD instead of GD. Empirically, it is found that SGD tends to favor feature learning [2].
>
> **Tests on other transformations.** To answer the referee, we ran additional experiments by measuring the relative sensitivity of predictors with respect to (i) 1-pixel translations and (ii) corruption of pixels at the boundary. We indeed confirm that results from (i) are qualitatively similar to applying small deformations, as we will explain in the final version of the manuscript. In case (ii), we expect that, if boundary pixels contain variance without conveying information about the class label, then the feature regime should be able to learn such an invariance, while the lazy regime would not. We confirm this to be the case for the FashionMNIST dataset (black and white clothes pictures).
>
> **On the definition of Rf.** Rf is a natural definition as it quantifies the relative smoothness of the predictor when moving in chosen directions of the input (e.g. small deformations) with respect to moving in random directions. In practice, this definition correlates most strongly with performance across different architectures  [3], which supports using it.  Yet, its absolute value is somewhat hard to interpret. A possible explanation for values of Rf larger than unity is that in the feature learning regime, the network can become insensitive to (uninformative) pixels on the edge of the image. Such effects lower the denominator in the definition of Rf, as noise on the edge pixels then does not affect the output. Achieving entirely quantitative predictions (e.g. explaining why the error curves for CIFAR-10 separate for large n while stability curves converge) would require a careful consideration of all the effects contributing to generalization, as stated in the text. Stability/sensitivity to deformation is one such effect, but there are others. For example, in the feature learning regime becoming insensitive to the pixels on the edge of the image may decrease sensitivity to Gaussian noise, so increase Rf overall. Becoming insensitive to high spatial frequencies (if they contain little information on the task) in the image may instead lower Rf, an effect that could be at play in the CIFAR10 data.
>
> **On the use of pooling in deep nets.** More recent architectures like ResNet and EfficientNets do not have pooling as part of the convolutional blocks as previous architectures like LeNet and VGG have.
>
> **Defining $\nu_t$ for real data.** This is an interesting suggestion. An attempt to measure $\nu_t$ in real data is present in [2]. That is done by considering the asymptotic decay of the eigenvalues coefficients of the Laplace kernel Gram matrix. A drawback of that approach is that all directions in space are treated equally, while presumably, the target function is smoother along some specific directions (e.g. smooth deformations), compared to random ones. An alternative approach is to take a state-of-the-art network trained on (e.g.) CIFAR10 and use it as a proxy for the target function. Indeed, such a predictor varies little along small deformations [3], consistent with a large $\nu_t$ in these directions.
>
> - [1] J. Lee, S. S. Schoenholz, J. Pennington, B. Adlam, L. Xiao, R. Novak, and J. Sohl-Dickstein. Finite versus infinite neural networks: an empirical study. arXiv preprint arXiv:2007.15801, 2020.
> - [2] S. Spigler, M. Geiger, and M. Wyart. Asymptotic learning curves of kernel methods: empirical data versus teacher-student paradigm. Journal of Statistical Mechanics: Theory and Experiment, 2020(12):124001.
> - [3] L. Petrini, A. Favero, M. Geiger, and M. Wyart. Relative stability toward diffeomorphisms indicates performance in deep nets. In Advances in Neural Information Processing Systems 2021.
> - [4] M. Geiger, S. Spigler, A. Jacot, and M. Wyart. Disentangling feature and lazy training in deep neural networks. Journal of Statistical Mechanics: Theory and Experiment, 2020(11):113301.

---

### Comment · Area_Chair_3BYQ · 2022-08-07
**Discussion period**

Thank you to all the reviewers for the great effort in reviewing the paper and the authors for the responses.

As in the discussion period, I want to ensure that reviewers have read the authors' responses and engage with the authors if needed.

If you haven't done this, could you please take a moment to read through the authors' responses, update the reviews to indicate that you have read the authors' responses, or communicate with the authors if needed? You can also share in private conversations with the reviewing team.

Please continue to share your thoughts. Thank you!

---

### Meta-Review · Area_Chair_3BYQ · 2022-08-28

**Recommendation:** Accept
**Confidence:** Certain

**Metareview:**

This paper studies different behaviors of feature learning and lazy training in the regression of Gaussian random functions and image classification. The main contribution includes insights into why feature learning in a particular setting is not robust (may overfit) compared to the lazy training regime for fully connected networks.

The reviewers pointed out several concerns about the presentation of the proofs, different behaviors of FCN and CNN, etc., which have been appropriately addressed by the authors. Overall, all reviewers appreciate the contribution of this paper, so I recommend accept.

**Award:**

Yes

---

### Decision · Program_Chairs · 2022-09-14

Accept